# Identifying tumor type and cell type-specific gene expression alterations in pediatric central nervous system tumors

Min Kyung Lee [1], Nasim Azizgolshani[1,2], Joshua A. Shapiro [3], Lananh N. Nguyen[4], Fred W. Kolling [5], George J. Zanazzi[5,6], Hildreth Robert Frost[7] & Brock C. Christensen [1,8,9] ✉

Central nervous system (CNS) tumors are the leading cause of pediatric cancer death, and these patients have an increased risk for developing secondary neoplasms. Due to the low prevalence of pediatric CNS tumors, major advances in targeted therapies have been lagging compared to other adult tumors. We collect single nuclei RNA-seq data from 84,700 nuclei of 35 pediatric CNS tumors and three non-tumoral pediatric brain tissues and characterize tumor heterogeneity and transcriptomic alterations. We distinguish cell subpopulations associated with specific tumor types including radial glial cells in ependymomas and oligodendrocyte precursor cells in astrocytomas. In tumors, we observe pathways important in neural stem cell-like populations, a cell type previously associated with therapy resistance. Lastly, we identify transcriptomic alterations among pediatric CNS tumor types compared to non-tumor tissues, while accounting for cell type effects on gene expression. Our results suggest potential tumor type and cell type-specific targets for pediatric CNS tumor treatment. Here we address current gaps in understanding single nuclei gene expression profiles of previously under-investigated tumor types and enhance current knowledge of gene expression profiles of single cells of various pediatric CNS tumors.

Central nervous system (CNS) tumors account for ~25% of pediatric cancer cases and are the leading cause of cancer death in children and adolescents in the United States[1]. However, CNS tumors in the pediatric population are rare in general, with 3.56 incidence per 100,000 for malignant tumors and 2.66 incidence per 100,000 for non-malignant tumors, which make these tumor types difficult to investigate[2]. Incident pediatric CNS tumors are comprised of many histologically distinct tumor types including pilocytic astrocytomas (15.2%), embryonal

tumors (9.4%), and neuronal/mixed neuronal-glial tumors (7.9%)[2]. Survival rates vary widely among tumor types, with a good 10-year survival of 95.4% for pilocytic astrocytomas and a poor 10-year survival of 15.9% for pediatric high-grade gliomas[2]. Pediatric CNS tumor patients are at risk of developing secondary neoplasms, with a 30-year cumulative incidence of malignant secondary neoplasms ranging from 4.7–7.8%[3,4]. The standard of care treatments for primary CNS tumors include surgery, radiotherapy, and chemotherapy with relatively

[1]Department of Epidemiology, Geisel School of Medicine at Dartmouth, Lebanon, NH, USA. [2]Department of Surgery, Columbia University Irving Medical Center, New York, NY, USA. [3]Childhood Cancer Data Lab, Alex's Lemonade Stand Foundation, Bala Cynwyd, PA, USA. [4]Department of Laboratory Medicine and Pathobiology, University of Toronto, Toronto, ON, Canada. [5]Dartmouth Cancer Center, Lebanon, NH, USA. [6]Department of Pathology and Laboratory Medicine, Geisel School of Medicine at Dartmouth, Lebanon, NH, USA. [7]Department of Biomedical Data Science, Geisel School of Medicine at Dartmouth, Lebanon, NH, USA. [8]Department of Molecular and Systems Biology, Geisel School of Medicine at Dartmouth, Lebanon, NH, USA. [9]Department of Community and Family Medicine, Geisel School of Medicine at Dartmouth, Lebanon, NH, USA. ✉e-mail: Brock.Christensen@dartmouth.edu

limited options for targeted therapy compared to tumors in other anatomic regions.

Recent advances in identifying molecular subtypes in various pediatric CNS tumor types have been made utilizing genomic, transcriptomic and epigenomic data as reflected in the 2021 World Health Organization classification of CNS tumors[5]. For example, medulloblastoma can be classified into four separate molecularly defined subtypes: WNT-activated, SHH-activated and *TP53*-wildtype, SHH-activated and *TP53*-mutant, and non-WNT/non-SHH[6–10]. In addition, supratentorial ependymoma can be categorized into *ZFTA* fusion-positive or *YAP1* fusion-positive[11,12]. A better understanding of the molecular variations that exist even among each tumor type has led to innovative treatment options. For example Larotrectinib and entrectinib, targeted therapies for *NTRK* fusion, which has been found in brain tumors, have been approved by the Food and Drug Administration to treat some brain tumors that are metastatic or unresectable with surgery[13,14].

In addition to the molecular characterization of bulk pediatric CNS tumor tissue, emerging work has begun to investigate the transcriptome and cellular states that exist in these tumors at the single-cell level. One of the first single-cell transcriptomics contributions focused on *H3K27M -altered* pediatric gliomas ($n = 6$, and 3300 cells) showed that tumors are mainly composed of progenitor cell-like oligodendrocyte populations, rather than differentiated malignant cells[15]. Later, Gojo et al. identified that cellular hierarchies in primary ependymomas ($n = 28$) reflect impaired neurodevelopment and that undifferentiated programs can infer prognosis[16]. Moreover, Gillen et al. revealed that subpopulations in ependymomas ($n = 26$) impact tumor molecular classification of bulk transcriptomes[17]. In medulloblastomas ($n = 25$ and 9000 cells), Hovestadt et al. identified specific subpopulations associated with molecular subtypes[7]. For example, Group 4 medulloblastoma are composed of differentiated neuronal-like neoplastic cells, while the other three groups are composed of subgroup-specific undifferentiated and differentiated neuronal-like malignant populations[7].

While these single cell and single nucleus transcriptomics studies in 85 total primary CNS tumors to date have improved our understanding of cell states in pediatric CNS tumors, there is still much to be investigated to advance optimal therapeutic options for both primary cancer treatment and reduction of secondary neoplasms. Due to limited sample availability for these rare pediatric CNS tumors, progress in single-cell-level characterization of these tumors has been relatively slow. Here, we characterized single nuclei gene expression profiles of 35 pediatric CNS tumors and 3 non-tumor pediatric brain tissues. Our study augments previous studies by incorporating single nuclei gene expression profiles of additional pediatric CNS tumor types (dysembryoplastic neuroepithelial tumors, gangliogliomas, etc.) and non-tumor pediatric brain tissue which have been limited in investigation to our knowledge.

In this study, we demonstrate the effects from cell type composition differences when comparing the transcriptome of pediatric CNS tumors and non-tumor pediatric brain tissue by integrating single nuclei RNA-seq and bulk tissue RNA-seq datasets as cell type heterogeneity is not considered when molecular profiles of bulk tumor tissue are compared to bulk non-tumor tissue.

## Results

Samples from pediatric central nervous system tumors and non-tumor pediatric brain tissue were obtained from patients being treated at Dartmouth Hitchcock Medical Center and Dartmouth Cancer Center from 1993 to 2017. Non-tumor pediatric brain tissues from the supratentorial regions were collected from patients undergoing surgical resection for epilepsy. Patient characteristics are described in Table 1. Pathological re-review for histopathologic tumor type and grade were done according to the 2021 World Health Organization CNS tumor

**Table 1 | Sample demographics**

| | Nontumor | Tumor |
|---|---|---|
| Sample size (N) | 3 | 35 |
| # of nuclei | 17,451 | 67,249 |
| Mean (Range) | Pooled | 1921.4 (234–5795) |
| **Age** | | |
| Mean (Range) | 6.2 (0.58–11) | 9.3 (0.75–18) |
| **Sex** | | |
| F | 2 (66.7) | 13 (37.1) |
| M | 1 (33.3) | 22 (62.9) |
| **Location** | | |
| Subtentorial | 0 (0.0) | 22 (62.9) |
| Supratentorial | 3 (100.0) | 13 (37.1) |
| **Tumor type** | | |
| Astrocytoma | | 8 (22.9) |
| Embryonal | | 6 (17.1) |
| Ependymoma | | 11 (31.4) |
| Glioneuronal/Neuronal | | 8 (22.9) |
| Glioblastoma | | 1 (2.9) |
| Schwannoma | | 1 (2.9) |
| **Grade** | | |
| Low (1 + 2) | | 20 (57.2) |
| High (3 + 4) | | 13 (37.1) |
| NEC/NOS | | 2 (5.7) |

classification system and categorized into the broader tumor types to balance sample size per tumor type[5]. Specific diagnoses for each sample can be found in Supplementary Table 1.

Additional molecular information of our pediatric CNS tumor cohort was determined from the bulk RNA-seq and DNA methylation data. Genetic variants were identified using bulk tissue RNA-seq data for all tumors except for two tumors due to low bulk RNA-seq data quality. Copy number variations (CNV) were determined using bisulfite treated DNA methylation array data. Genetic and cytogenic variations varied among tumors and tumor types (Supplementary Fig. 1, Supplementary Data 1). Many of the somatic genetic variants (insertions, SNPs, deletions, etc.) detected within the pediatric CNS tumors were identified in genes known to have functions in epigenetic processes. For example, almost half of the tumors (14/33), across tumor types, had genetic variants in HIST1H1E, a H1.4 linker histone gene. Interestingly, across all but one tumor sample, tumors had genetic variants in MALAT1, a non-coding RNA with roles in nuclear organization and modulation of gene expression. While the variants were filtered under more stringent cut-offs, there may have been undetected variants or misclassified variants as they were called from bulk RNA-seq in comparison to a reference. CNV patterns in some tumor types were as expected from previous literature. For instance, 5 out of the 9 ependymoma had chromosome 1q gain, which has been considered to be an early tumorigenic event in ependymoma[18,19].

### Integrated de-multiplexing method to increase single nuclei RNA-seq data yield

Using lipid-tagged hashtag oligonucleotides (HTO), 34 samples (out of 38 total samples) were multiplexed in 17 pools to collect 10X genomics snRNA-seq data[20]. The distribution of samples across sequencing runs and pools is provided in Supplementary Table 1B. As the level of HTO tagging was often insufficient to distinguish from background tag sequence levels to be efficiently demultiplexed in downstream analyses, we aimed to augment demultiplexing by analyzing bulk RNA-seq derived genotype data from each nucleus together with HTO information and assign additional nuclei to specific samples (Fig. 1A). The

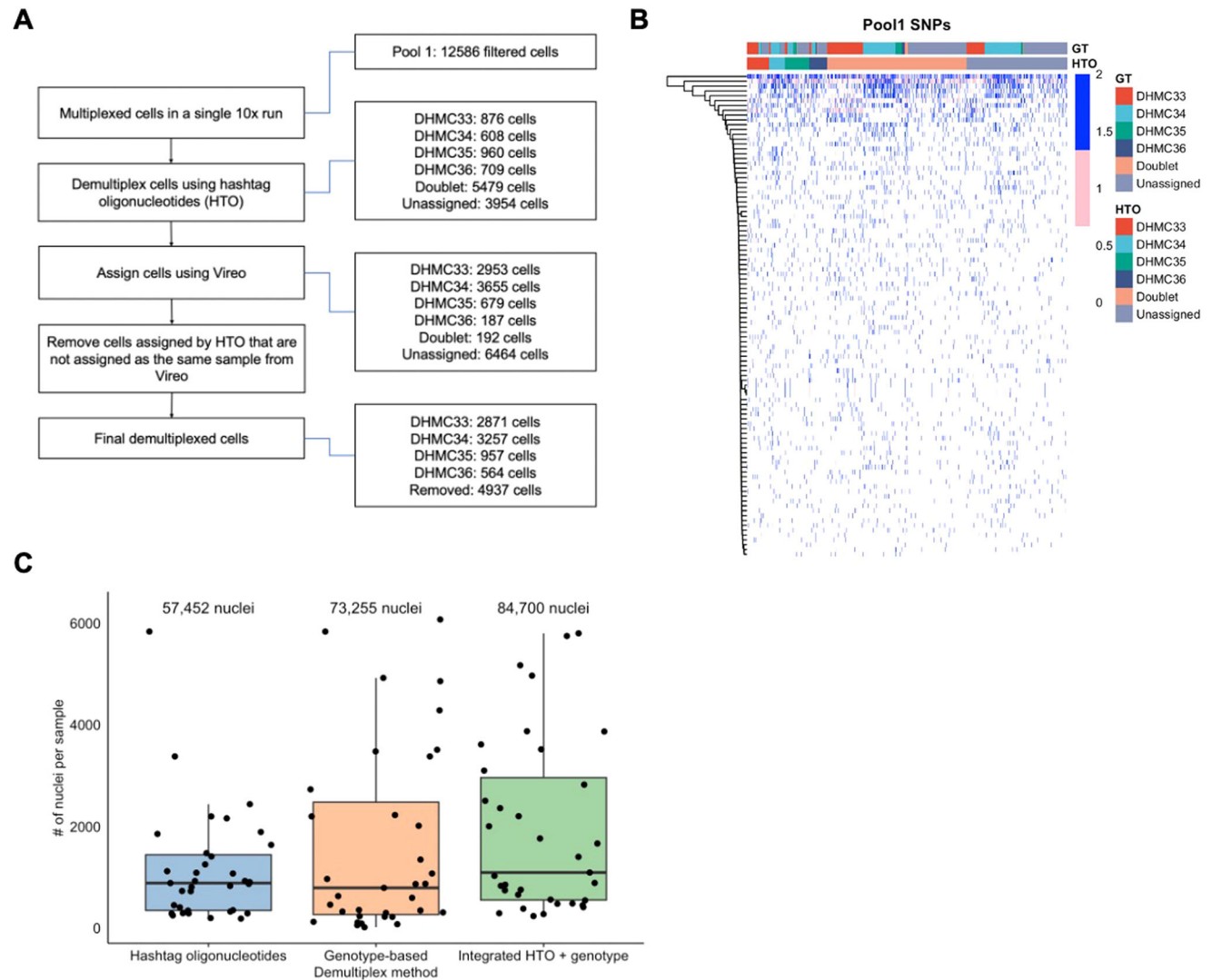

**Fig. 1 | Integrative method to demultiplex pooled samples increases nuclei per sample from single-nuclear RNA-seq data. A** Diagram of integrated method for demultiplexing pooled samples. Multiplexed samples were first demultiplexed using hashtag oligonucleotide (HTO) counts. Cells assigned using HTO were filtered for those that did not match the sample assignment from genotype-based method. Cells unable to be assigned to a sample from HTO were assigned based on genotype information. On the right are the number of cells retained at each step of the integrated demultiplex method for Pool #1. **B** Example of genotype information (Pool #1) used to demultiplex samples. Blue indicates 100% alternate allele presence. Pink indicates heterogeneous alternate allele presence. White indicates no alternate allele depth presence. Tracking bars indicate the samples assigned based on HTO or genotype (GT). **C** Distribution of the number of nuclei assigned per sample based on hashtag oligonucleotides, genotype-based method, or integrated method. Each point indicates one sample. The total number of nuclei obtained for each method is labeled on the top of the boxplot. In the boxplot, the low ends of the segment indicate the minimum and the high ends of the segment indicate the maximum. Lower bounds of the box indicate the 25th percentile and the higher bounds of the box indicate the 75th percentile. Segment in the middle is the median. Source data are provided as a Source Data file.

large proportion of doublet assignment from the HTO method in contrast to the much smaller number of doublet assignments from genotype data suggests the oligonucleotides were not stained as effectively as would have been preferred (Fig. 1B). From the similarities in the genotype profiles in samples that were assigned to a specific sample from both methods compared to the ones labeled as doublet from HTO, but as the same sample through the genotype method provided more confidence on the accuracy of the sample assignment in even the nuclei that the HTO method had considered to be a doublet.

To summarize our demultiplexing process, we first used HTOs to assign the nuclei to their respective samples. For samples that were assigned confidently with the HTO, we filtered to keep only the nuclei that were assigned to the same sample concordantly using genotype information. For nuclei that were either unassigned to a sample or assigned as a doublet with HTO, we assigned nuclei to samples using genotype information (detailed in the methods section). The final set of nuclei per sample were comprised of the filtered nuclei from HTO and genotype identified nuclei. An example of the single nucleotide variants identified per pool along with their assigned sample can be found in Fig. 1B. An example of how many nuclei were obtained for one pool, during each step, is shown in Fig. 1A on the right. The integrated demultiplex method classified an average of 1921 nuclei per sample (range = 234–5795, Table 1). The number used in downstream analysis per sample is included in Supplementary Table 1. The total number of demultiplexed nuclei was increased 47.4% (additional 27,248 nuclei) using the integrated approach over the HTO-only method, and 15.6% (additional 11,445 nuclei) over the genotype-based method alone (Fig. 1C). Gene expression profiles for a total of 84,700 nuclei were used for downstream analyses.

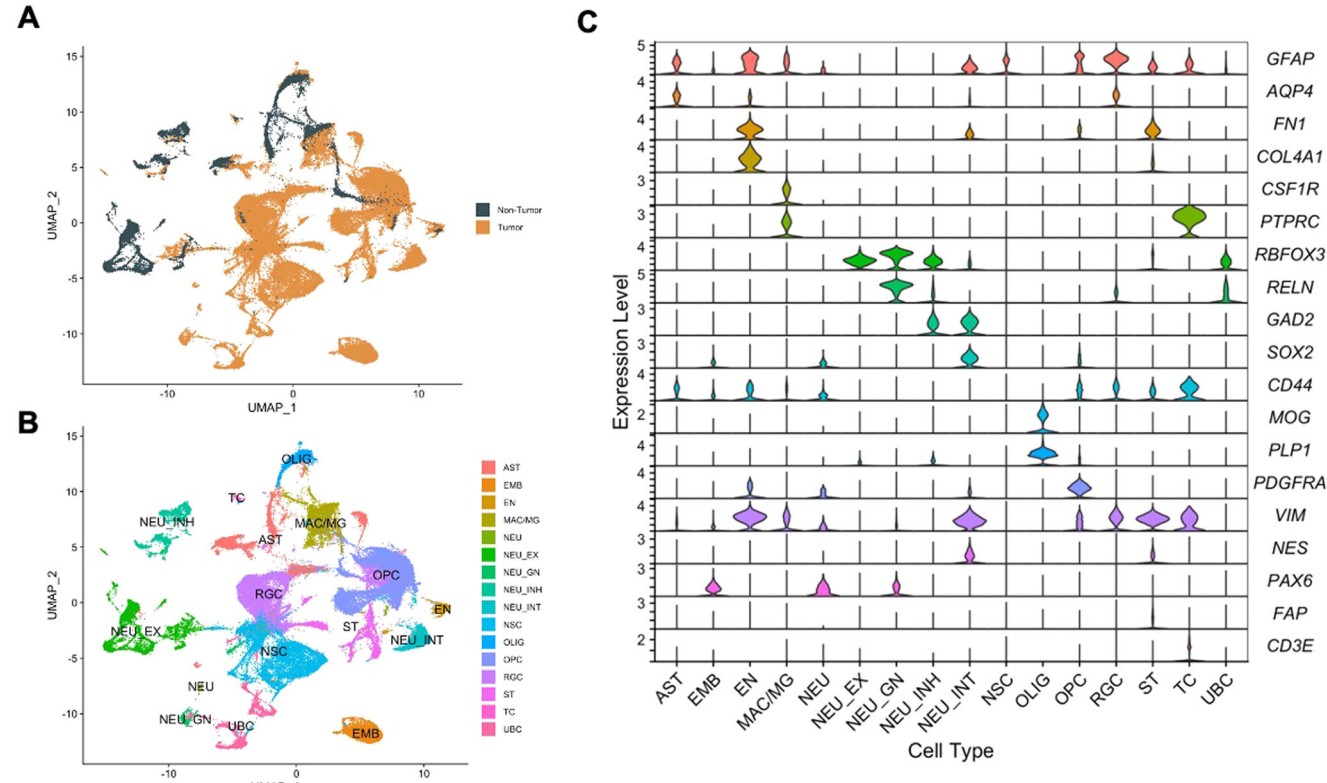

**Fig. 2 | Heterogeneity of cell types in pediatric CNS tumor tissue and non-tumor pediatric brain tissue. A** Uniform Manifold Approximation and Projection (UMAP) clustering of the 84,700 nuclei colored based on tissue. Dark green indicates nuclei from non-tumor tissue. Orange indicates nuclei from tumor tissue. **B** UMAP of the 84,700 nuclei colored by major cell type. **C** Violin plots of gene expression levels of classical gene markers for cell types present in the brain by major cell type cluster. Astrocytes (AST): *GFAP* and *AQP4*; Endothelial cells (EN): *FN1* and *COL4A1*; Macrophage/microglia (MAC/MG): *CSF1R* and; Neurons and unipolar brush cells (NEU, NEU_EX, NEU_GN, UBC): *RBFOX3* and *RELN*; Inhibitory neurons and interneurons (NEU_INH, NEU_INT): *GAD2*; Neural stem cells (NSC): *SOX2* and *CD44*; Oligodendrocytes (OLIG): *MOG* and *PLP1*; Oligodendrocyte precursor cells (OPC): *PDGFRA*; Radial glial cells (RGC): *VIM, NES*, and *PAX6*; Stromal cells (ST): *FAP*; T cells (TC): *CD3E*. Source data are provided as a Source Data file.

## Cell-type heterogeneity in pediatric central nervous system tumors and non-tumor pediatric brains

Out of 84,700 nuclei, 67,249 nuclei (79%) were from pediatric CNS tumors and 17,451 nuclei (21%) were from non-tumor tissue (Fig. 2A). We utilized unsupervised clustering from components of uniform manifold approximation and projection (UMAP), a dimension reduction strategy, to identify nuclei most similar to each other based on gene expression levels. Across all samples, snRNA-seq data revealed 58 clusters that were grouped into 16 major cell types: astrocytes (AST), embryonal tumor cells (EMB), endothelial cells (EN), macrophage/microglia (MAC/MG), neurons (NEU), excitatory neurons (NEU_EX), granular neurons (NEU_GN), inhibitory neurons (NEU_INH), interneurons (NEU_INT), neural stem cells (NSC), oligodendrocytes (OLIG), oligodendrocyte precursor cells (OPC), radial glial cells (RGC), stromal cells (ST), T cells (TC), and unipolar brush cells (UBC) (Supplementary Fig. 2A, Fig. 2B). The clusters were classified into cell types using classical markers for cell types found in the brain (Table 2). The following gene markers for cell types were used: *GFAP* and *AQP4* for astrocytes; *FN1* and *COL4A1* for endothelial cells; *CSF1R* and *PTPRC* for macrophage/microglia; *RBFOX3* and *RELN* for neurons and unipolar brush cells; *GAD2* for inhibitory neurons and interneurons; *SOX2* and *CD44* for neural stem-like cells; *MOG* and *PLP1* for oligodendrocytes; *PDGFRA* for oligodendrocyte precursor cells; *VIM, NES*, and *PAX6* for radial glial cells; *FAP* for stromal cells; *CD3E* for T cells. Not all gene markers corresponded to expected expression levels for the major cell types. For example, the neural stem cells (NSCs) did not express classical neural stem cell-like genes (*SOX2* and *CD44*) but were

identified by enrichment testing of neural stem cell/neural progenitor-like cell gene sets. Nuclei in the EMB clusters (EMB1, EMB2) did not have detectable levels of currently known markers for classical brain cell types. These nuclei were found to be specific to embryonal tumors and clustered distinctly from other tumor types; therefore, were named the EMB (embryonal) cluster. These marker-based cell type classifications were subsequently validated by enrichment of cell type-specific pathways using the Variance-adjusted Mahalanobis method, a single-cell-level pathway enrichment method (Fig. 2C, Supplementary Fig. 2B)[21–28]. The cell

### Table 2 | Classical markers for cell types

| Cell type | Markers |
|---|---|
| Astrocytes | *GFAP; AQP4* |
| Endothelial cells | *FN1; COL4A1* |
| Macrophage/Microglia | *CSF1R; PTPRC* |
| Neurons/Unipolar brush cells | *RBFOX3; RELN* |
| Inhibitory neurons/Interneurons | *GAD2* |
| Neural stem-like cells | *SOX2; CD44* |
| Oligodendrocytes | *MOG; PLP1* |
| Oligodendrocyte precursor cells | *PDGFRA* |
| Radial glial cells | *VIM; NES; PAX6* |
| Stromal cells | *FAP* |
| T cells | *CD3E* |

Italics are necessary as they are gene names.

type-specific pathways used for enrichment testing were derived from single-cell RNA-seq experiments of developing human and mouse brains.

We compared the distribution of cell type proportion from publicly available single-cell RNA-seq datasets of ependymoma (data accessible at NCBI GEO database, accession GSE141460, $n = 6$)[16] and medulloblastoma (data accessible at NCBI GEO database, accession GSE214357, $n = 6$)[29] to our cohort of ependymoma and medulloblastoma subjects to validate our observations due to the smaller sample size of our cohort. We assessed the two most prevalent cell types in each tumor type: NSC and RGC in ependymoma; NSC and UBC in medulloblastoma. The proportions of NSC and RGC in ependymoma between our cohort and those in GSE141460 were not significantly different (NSC: $p = 0.46$, RGC: $p = 0.3$; Supplementary Fig. 3). Similarly, the proportions of NSC and UBC between medulloblastoma between our cohort and those in GSE214357 were not statistically different (NSC: $p = 1$, UBC: $p = 0.9$). Our results suggest that cell type distributions we observed in pediatric CNS tumor subtypes are consistent with independent pediatric CNS tumors of matched subtype.

As cellular stemness is a key characteristic in sustaining tumor survival and malignancy[30,31], we set out to investigate stemness in each cell types of our pediatric CNS tumors. To identify stem-like phenotypes in our tumor nuclei population, we investigated the expression levels of classically used markers of cancer stem cells (*ITGA6, CD44, PROM1, NES, MSI1, MYC, NANOG, SOX1, SOX2, POU5F1, VIM, SDC1, SDC2, GPC1, GPC2*), as well as an enrichment score for stemness from Tirosh et al.[32,33]. Although expression in some stemness marker genes were not detected, in those that were detected, levels of expression for genes classically used for to isolate stem-like cells in literature varied among the different cell types (Supplementary Fig. 4, Supplementary Table 2). Interestingly, cell types expected to be more differentiated, like astrocytes, had relatively high levels of CD44 and VIM, and these genes were expressed in many of the cell types. As there were some genes that failed to be detected in a lot of the cell types, we calculated stemness scores based on a larger set of stemness related genes. All cell types except for neuronal cells had high levels of stemness. Although the NSC-like cluster had a high stemness score, the expression of cancer stem cell markers was minimal. Unexpectedly, the UBC-like clusters also had elevated stemness scores. While gene expression levels may not always correlate with protein expression, our results indicate cell types identified using classical stem cell markers may not capture all tumor cells with stemness features.

Next, we tested for potential associations of clinical variables with tumor stemness scores. We first assessed the distribution of stemness scores among nuclei in each sample and determined the median stemness score (Supplementary Fig. 5). We found that the stemness scores were higher in embryonal compared to other tumor types or non-tumor tissue (Supplementary Fig. 6A). Specifically, embryonal tumors had significantly higher stemness scores compared to astrocytomas ($p = 0.03$), ependymoma ($p = 0.02$), and glioneuronal/neuronal tumors ($p = 0.029$). Compared with low grade tumors, high-grade tumors had higher stemness scores ($p = 0.008$, Supplementary Fig. 6B), and somewhat unexpectedly, stemness score was positively correlated with age ($R = 0.47$, $p = 0.004$, Supplementary Fig. 6C). When this association was investigated by each tumor type, only astrocytoma was found to be statistically significant ($R = 0.72$, $p = 0.043$, Supplementary Fig. 6D). However, positive correlation coefficients were observed for the other tumor types, and sample size limited statistical power. No difference in stemness score was observed between tumors in the subtentorial and supratentorial regions of the brain ($p = 0.600$, Supplementary Fig. 6E). Our results indicate that stemness level of single cells is associated with tumor type and grade, which may be important when considering potential for therapy resistance and metastasis and when developing targeted therapies.

To reveal any specific cell populations that are only present in a restricted set of tumor types, we evaluated the association between cell type proportions and tumor type (Fig. 3, Supplementary Fig. 7). Non-tumor tissue contained nuclei from all major expected cell types found in normal brain, including astrocytes, oligodendrocytes, and excitatory and inhibitory neurons which demonstrated the high-quality data derived from the non-tumor tissues. Some tumor samples had small proportions of cell types normally present only in non-tumor tissue, such as excitatory neuron cluster #5 (NEU_EX5) and inhibitory neuron cluster #2 (NEU_INH2). These cases are likely the result from the inclusion of cells from the tumor margin. Non-tumor tissues had limited numbers of nuclei from progenitor-like cell types, like NSCs, RGCs, or UBCs. While OPCs are a progenitor cell type, they are also found in normal brain tissue. The non-tumor OPCs were limited to the OPC4, a population transcriptionally distinct from tumor OPCs residing in OPC1-3.

Some cell types were exclusive to a specific tumor. For example, the glioblastoma sample was comprised of 91% NSC1, and an ependymoma sample consisted predominantly (86%) of OPC2. MG2 was present at higher proportions (mean = 3.3%, range = 0.3–31.2%) in tumors compared to non-tumor tissue (0.9%). All astrocytomas had at least small proportions of A4, OPC1, and OPC5. The embryonal tumors had cell types that were more neuronal (apart from EMB cell types) like NSCs and UBCs. Large proportions of ependymoma samples were made of RGC clusters. The glioneuronal/neuronal tumor type samples were more varied in terms of which cell types were more present in each tumor. The expanded cell types were consistent with some known cell types of origin for these tumors, such as the RGCs in the ependymomas.

## Cell-type-specific pathway enrichment in pediatric CNS tumors

We set out to investigate potential therapeutically targetable pathways specific to cell types associated with tumorigenesis and progression in the pediatric CNS tumors. First to determine cell type-specific pathway enrichment in each nuclei of the tumor samples, we conducted a pathways analysis at the single-cell level using the Variance-adjusted Mahalanobis (VAM) method, which computes cell-level pathway scores that account for the technical noise and inflated zero counts of single-cell RNA-seq data[21]. We used 196 pathways from the MSigDB Pathway Interaction Database (PID) collection for our enrichment testing[23,24,34]. The cell-level enrichment p-values generated by VAM were corrected for false discovery rate using the Benjamini-Hochberg method and classified to be significantly enriched in each nucleus if the FDR adjusted p-value was less than 0.1 as binary classifications (enriched or not enriched).

Next, we determined any pathways that were more specific for each cell type by comparing the enrichment classifications from above for each cell type to determine important cell type-specific pathways. The PID pathways were considered to be important/specific to the cell type under adjusted *p*-value < 0.05 threshold in the differential enrichment test. For cell types with a limited presence in tumor tissues, like many of the excitatory neurons and A1, we observed no pathways that were specific to the clusters (Supplementary Fig. 8A, Supplementary Data 2). The immune-related cells (MG1, MG2, and TC), which were present in tumor tissue at slightly higher levels than in non-tumor tissue, had more than 44% of the PID pathways specific to these cell types. The high percentage of PID pathways that were important in the immune-related cell types is likely due to the relatively greater number of cytokine and other immune-associated pathways are included in the PID database.

All NSC clusters, except for NSC6 (3.6% pathways), had more than 10% of PID pathways that were important to the NSCs (range = 11.73–42.35, Supplementary Fig. 8A, Supplementary Data 2). While there were no shared pathways that were considered to be important

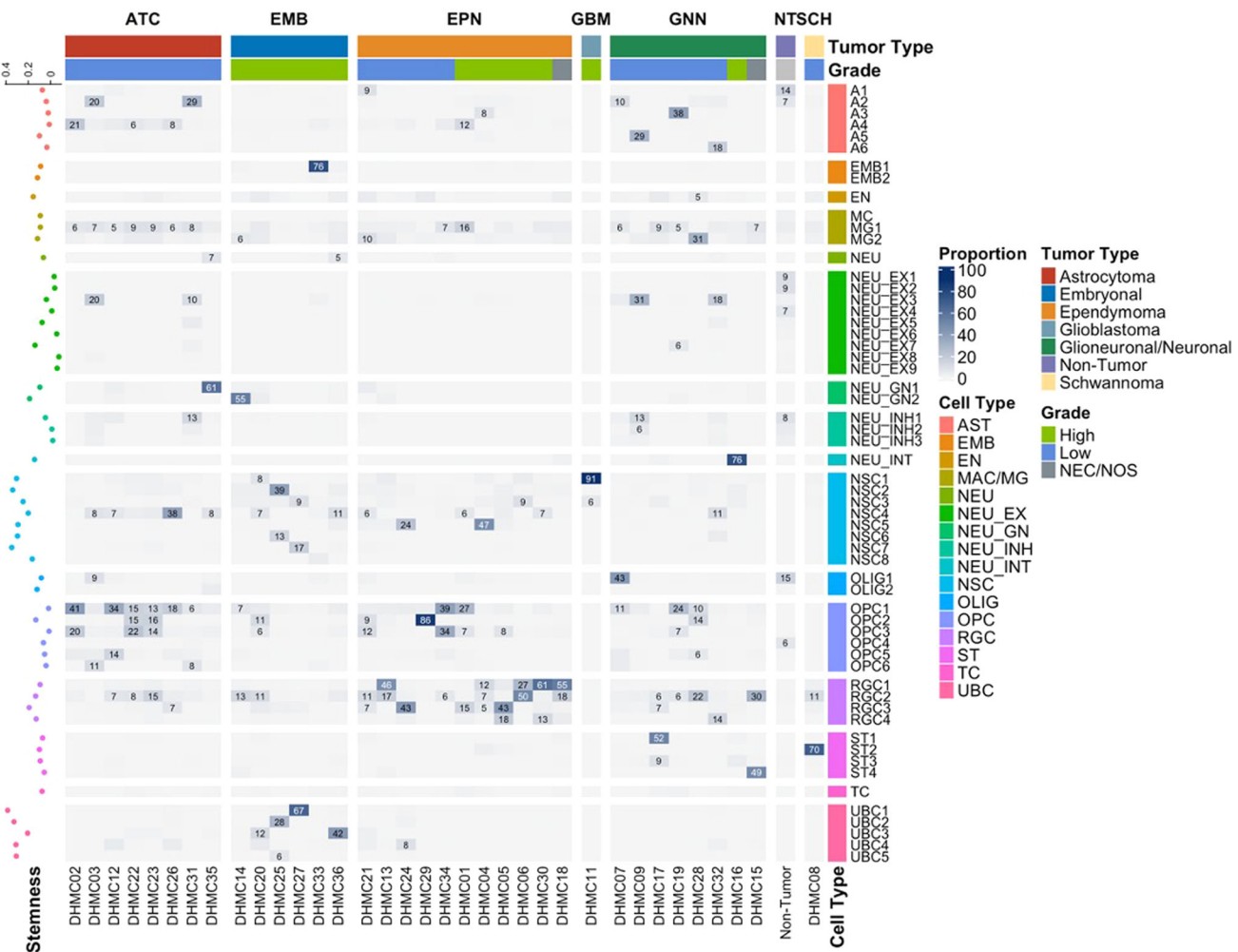

**Fig. 3 | Tumor type-specific presence of cell types.** Heatmap of the proportions (%) of each cell type present in each sample. Scatter plot on the left of the heatmap indicates the median stemness level of each cell type. Horizontal tracking bars indicate the tumor type and grade of each sample. Vertical tracking bars indicate the major cell types of the nuclei. The cell types with greater than 5% are labeled within each cell. ATC astrocytoma, EMB embryonal tumors, EPN ependymoma, GBM glioblastoma, GNN glioneuronal/neuronal tumors, NT non-tumor, SCH schwannoma. Source data are provided as a Source Data file.

in all 8 NSC clusters, there were numerous pathways shared among majority of the NSCs (Fig. 4A, Supplementary Fig. 8B). The retinoic acid pathway and telomerase pathway were considered to be important in 7 of 8 NSC clusters (Fig. 4B). Aurora-B, PLK1, FOXM1, E2, ATR, FOXO, Retinoic Acid pathways were considered to be important in just 6 of the 8 NSC clusters. Our results provided potential cell type-specific targets within these PID pathways important for each cluster for future therapeutic strategies.

**Transcriptomic alterations in tumors compared to non-tumor at the single-cell level**

We next aimed to determine transcriptomic alterations in pediatric CNS tumors compared to non-tumor pediatric brain tissue. In bulk differential gene expression analyses, it is typically not possible to account for the impact of cell composition differences on gene expression levels[35–38]. Here, using single nuclei level data, we compared expression of the 4000 most variable genes in nuclei from each tumor type to the gene expression of nuclei in non-tumor tissue, controlling for cell-type composition differences (Fig. 5A). Genes were considered differentially expressed if they met the FDR < 0.05 threshold.

As expected, adjusting for cell type proportions reduced the number of significantly differentially expressed genes compared with cell-type-unadjusted analyses. However, importantly, cell-type-adjusted analyses identified on average 200 genes per tumor type

that not observed in unadjusted models. (Fig. 5B, Supplementary Fig. 9A, B, Supplementary Data 3). Moreover, cell type-unadjusted analyses identified average 758 genes per tumor types that were not detected in adjusted models. Genes uniquely identified in cell-type-adjusted models represent underlying tumor biology that was obscured by variation in cell type proportions composing the tumor microenvironment across subjects (Fig. 5C). For example, *WNT3A*, a gene shown to mediate glioblastoma progression[39] was shown to be upregulated in glioneuronal/neuronal tumors and Schwannoma only using the adjusted analysis (Supplementary Data 3). Furthermore, the unadjusted model often gave estimates that were contrary to the direction of change from the adjusted model. For example, *FAT2* was significantly decreased (estimate = −1.80) in embryonal tumors relative to non-tumor tissue in the adjusted model but significantly increased (estimate = 0.42) in embryonal tumors in the unadjusted model. Also, *FGFR2* had significantly increased in expression (estimate = 0.76) in the Schwannoma nuclei relative to non-tumor tissue in the adjusted model but was significantly decreased in expression (estimate = −0.52) in the unadjusted model.

Using cell type-adjusted models, we detected tumor type-specific alterations in gene expression compared to non-tumor tissue. In astrocytomas, we identified 958 significantly downregulated and 970 significantly upregulated genes compared to non-tumor tissue (FDR < 0.05). Genes upregulated in astrocytomas include *ID4, CD74*

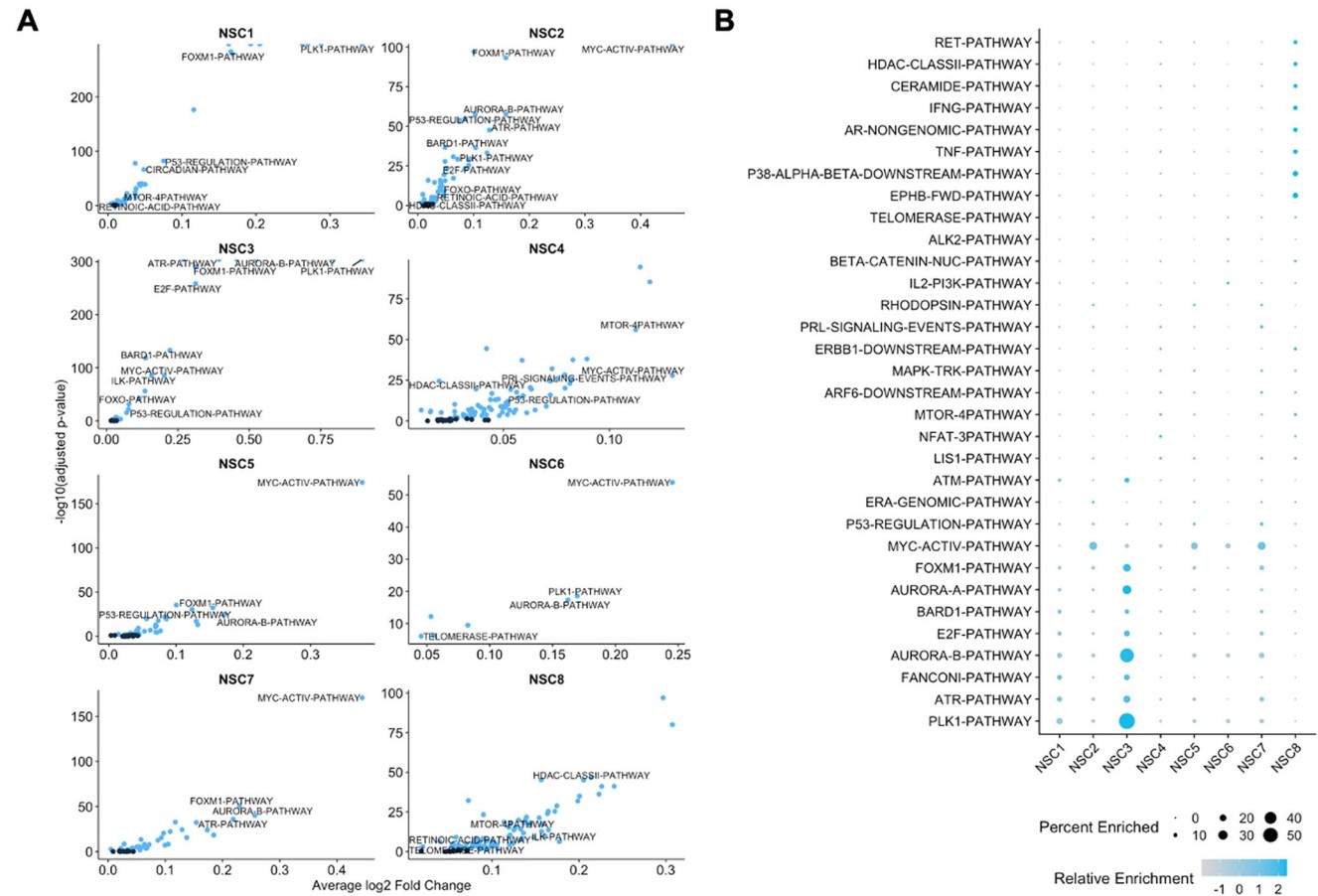

**Fig. 4 | Enriched pathways in neural stem cell-like cells in pediatric CNS tumors.**
**A** Differentially enriched pathways from Pathways Interaction Database (PID) in the NSC subpopulations compared to all other cell clusters in pediatric CNS tumors as identified by Wilcoxon rank sum tests. Blue points indicate statistically significantly enriched pathways at adjusted *P*-value threshold of 0.05. Labeled pathways indicate more commonly enriched pathways in the NSC subpopulations. Average log2 Fold Change indicates the fold change of # of cells with enriched pathways in NSCs

compared to all other cell types. The few points that appear to be cut-off have −log10(adjusted *P*-value) of infinity as the adjusted *P*-values were essentially zero. **B** Relative enrichment and percentage expressed in cluster of the top enriched pathways per NSC clusters. Color indicates relative enrichment. Size indicates percentage expressed in each NSC cluster. Source data are provided as a Source Data file.

and *FOS*. The differentially expressed (DE) genes in astrocytomas were associated with translation-related and nonsense-mediated decay-related processes (Supplementary Fig. 10A, Supplementary Data 4). Embryonal tumors had 915 downregulated and 944 upregulated genes relative to nontumor tissue that were associated with rRNA processing and translation-associated processes (Supplementary Data 5, Supplementary Data 5). In embryonal tumors, the topmost DE genes included many ribosome-associated genes like *RPS2, RPLP1*, and *RPL13A* as well as histone H3.3 related genes like *H3F3A* and *H3F3B*. Ependymomas had 1024 downregulated and 1213 upregulated genes compared to non-tumor tissue. The topmost DE genes were *IGFBP5*, *CFAP54* and *COLEC12*. Similar to astrocytomas, DE genes in ependymomas were associated with translation and nonsense-mediated decay-related processes (Supplementary Fig. 10C, Supplementary Data 6). Glioneuronal/neuronal tumors had 1,035 downregulated and 1,079 upregulated genes relative to non-tumor tissue; these genes that were associated with extracellular matrix and integrin-related processes and MET signaling (Supplementary Fig. 10D, Supplementary Data 7). *TAFA1, ALK*, and *VAV3* were some of the topmost DE genes in glioneuronal/neuronal tumors. In the glioblastoma, there were 1,575 downregulated genes and 524 upregulated genes that were associated with RNA processing and translation-related processes (Supplementary Fig. 10E, Supplementary Data 8). Some genes that were topmost DE in glioblastoma nuclei include *RMST, ID4* and *PBX3*. Lastly, in the

Schwannoma, there were 864 downregulated genes and 813 upregulated genes relative to non-tumor tissue that were associated with elastic fibers and RHO/RAC1 GTPases cycles (Supplementary Fig. 10F, Supplementary Data 9). *CEMIP, THSD4*, and *GPC6* were among the topmost DE genes in the Schwannoma nuclei. The list of differentially expressed genes and their associated pathways per tumor type are listed in Supplementary Data 3–9.

Of the 4000 most variable genes that were used in differential gene expression analysis, there were 558 genes that were differentially expressed in all six of the tumor types, 717 in five of the tumor types, and 596 in four of the tumor types compared with non-tumor tissue (Fig. 6A and Table 3). There were differentially expressed genes specific to a single tumor type: 43 genes for astrocytomas, 61 for embryonal tumors, 52 for ependymomas, 68 for glioneuronal/neuronal tumors, 98 for glioblastoma, and 57 for Schwannoma. While 60.9% (340/558) of the differentially expressed genes shared among all the tumor types were either increased or decreased the same direction, the remainder of genes varied in the direction of change based on tumor type compared to non-tumor tissue. The proportion of genes that either increased or decreased in the same direction for the shared significantly differentially expressed among all tumor types were significantly higher than expected ($p < 2.2 \times 10^{-16}$). Protein-coding genes with increased expression across all tumor types included *E2F7, ETS1, EZH2, ID3/4, MKI67, PIK3R3*, and *TOP2A*.

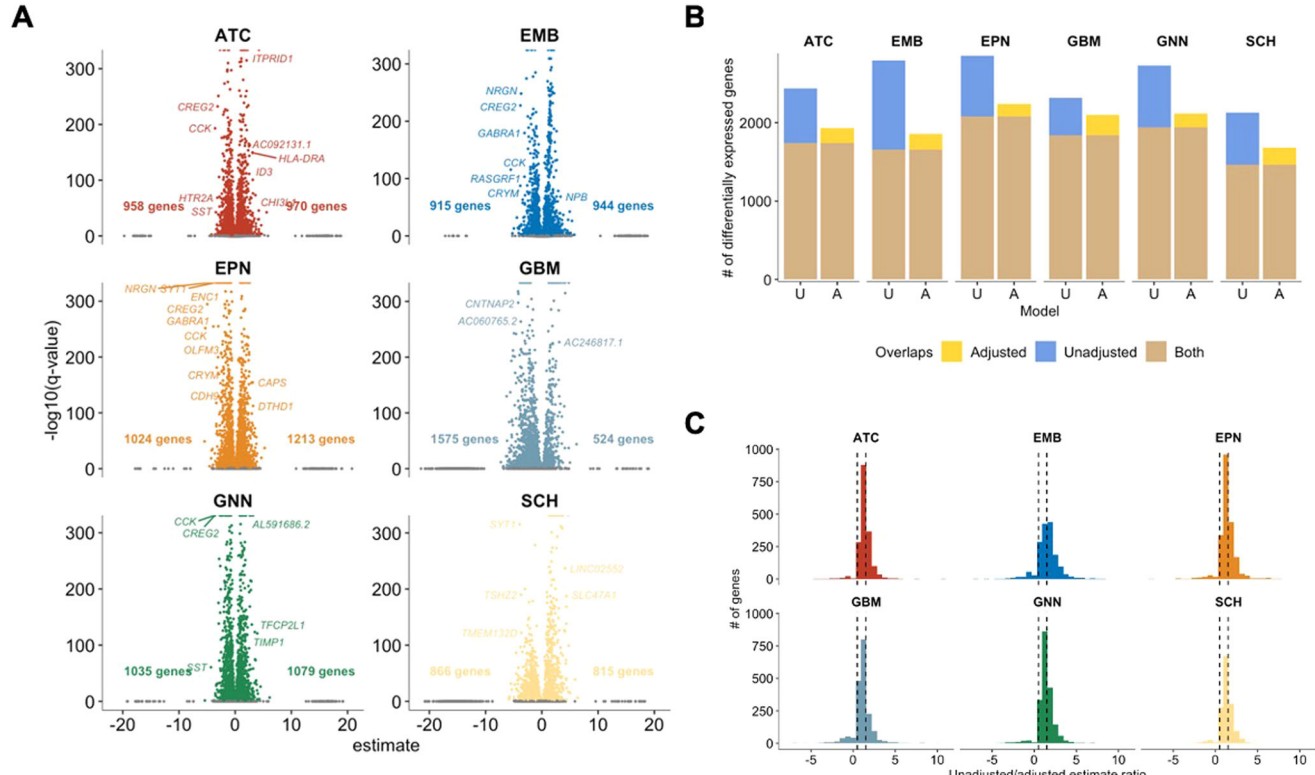

**Fig. 5 | Adjusting for cell type identity identifies previously unlinked genes associated with pediatric CNS tumor types. A** Volcano plot of differentially expressed genes for each tumor type compared to non-tumor tissue, adjusted for major cell type. Number of genes on the left of the volcano plot indicate genes that are downregulated compared to non-tumor tissue. Number of genes on the right of the plot indicate genes that are upregulated compared to non-tumor tissue. Some examples increased or decreased in each tumor type compared to non-tumor tissues are labeled. **B** Comparison of the number of differentially expressed genes in the adjusted model and the unadjusted model per each tumor type. Brown bars indicate the number of differentially expressed genes identified by both unadjusted and adjusted model. Blue bars indicate the number of differentially expressed genes in only the unadjusted models. Yellow bars indicate the number of differentially expressed genes in only the adjusted models. **C** Distribution of differential expression estimates in the unadjusted model compared to estimates in adjusted model per tumor type. Dashed lines at 0.5 and 1.5 to indicate genes with similar estimates in the two models. ATC astrocytoma, EMB embryonal tumors, EPN ependymoma, GBM glioblastoma, GNN glioneuronal/neuronal tumors, SCH schwannoma. Source data are provided as a Source Data file.

We conducted a pathways analysis of the genes with increased expression across all tumor types, and genes with decreased expression across all tumor types with Reactome pathways to understand the molecular context of the differentially expressed genes[40]. Interestingly, translation or nonsense-mediated decay-related processes having increased expression across all tumor types compared to non-tumor tissues (Fig. 6B). Shared decreased protein-coding genes across all tumor types included *FOXP2, GABRA1/2/4/5, NRGN, SST*, and *SYNPR*. Even when differential gene expression analyses were adjusted for cell type, across all tumor types, there was decreased expression in genes associated with neuronal system such as transmission across chemical synapses and activation of NMDA or GABA receptors (Fig. 6C). Hierarchical clustering of the differentially expressed genes revealed that transcriptomic alterations were similar in ependymomas and glioneuronal/neuronal tumors and likewise in astrocytomas and embryonal tumors (Fig. 6A).

## Discussion

In this study, we characterized gene expression profiles of 84,700 nuclei from snRNA-seq of 35 pediatric CNS tumors and 3 pediatric non-tumor brain tissues. We utilized an integrated hashtag oligonucleotide and genotype-based methods to maximize the number of sample-assigned nuclei from our multiplexed snRNA-seq experiment. Although the original MULTI-seq[20] work showed that multiplexing nuclei was feasible, some difficulty encountered with the approach in our study may have been attributable to use of fresh frozen samples

that had been stored in the freezer for decades. In our study, we detail a unique approach to increase the number of cells assigned to a specific sample from pooled sequencing runs by integrating a genotype-based approach to demultiplex snRNA-seq data. We included demultiplexed nuclei in alignment by both methods as well as demultiplexed nuclei by either method if the other method had an undetermined assignment to increase the number of nuclei for this analysis. As some nuclei were not in alignment for both, limitations in accuracy of sample assignment remain in our downstream analysis. Future studies are expected to benefit from our integrated demultiplexing method to maximize data usage while decreasing the cost of snRNA-seq experiments.

Due to the rarity of pediatric CNS tumors, it is difficult to accrue a large sample size for a highly powered study. Although our sample size was limited, our study incorporates pediatric CNS tumor types that have been limited in characterization with single cell or single nuclei RNA-seq such as gangliogliomas and further enhances the current limited literature on some pediatric CNS tumor types. Moreover, our dataset incorporates a large number of nuclei per sample with an average of around 1920 nuclei, which is a relatively larger nuclei sample size per sample in comparison to some other studies. We also incorporated non-tumor pediatric tissues in our experiment, which to our knowledge have been understudied in previous pediatric CNS tumor single-cell RNA-seq studies. Lastly, our cohort of pediatric CNS tumor patients include longitudinal follow up data, from which we were able to assess five-year recurrence for. We describe changes in cell type proportions specific to each tumor type and use this

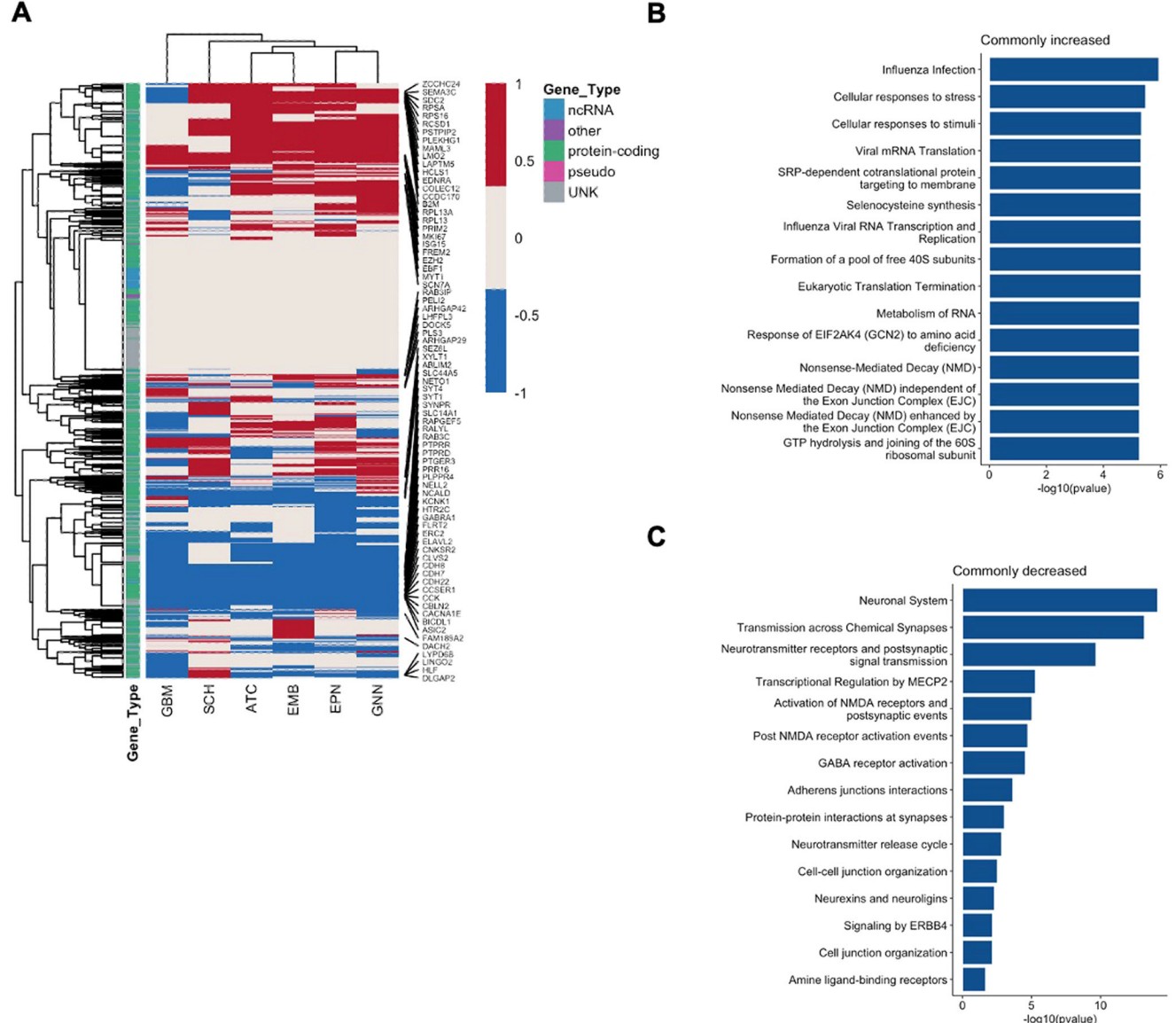

**Fig. 6 | Transcriptomic alterations in pediatric CNS tumor cells compared to nontumor pediatric brain cells. A** Heatmap of differential expression direction and significance in all 4000 genes tested in differential expression analyses. Red indicates significantly upregulated in the tumor type compared to non-tumor tissue. Blue indicates significantly downregulated in the tumor type compared to non-tumor tissue. Gray indicates the gene is not significantly differentially expressed. Tracking bar indicates the gene type: non-coding RNA (ncRNA), protein-coding, pseudo-gene, unknown (UNK) or other gene type. **B** Top Reactome pathways associated with genes commonly upregulated across all tumor types. **C** Top Reactome pathways associated with genes commonly downregulated across all tumor types. Enrichment of each pathway is tested using hypergeometric models. Source data are provided as a Source Data file.

information to identify the gene expression profiles and pathways enriched across tumor and normal samples through a cell type-adjusted analysis. Extended studies with larger sample size in collaboration with other medical centers or consortiums are needed to increase statistical power.

We characterized major cell subpopulations in specific tumor types, some of which have been limited in investigation. This includes the expansion of oligodendrocyte precursor cell (OPC) subpopulations in astrocytomas, and unipolar brush-like cells (UBC) with high stemness levels enriched in embryonal tumors. In the ependymomas, there was a significant presence of radial glial-like cells (RGC). Some glioneuronal/neuronal tumors featured stromal cells (ST) that were less present in other tumor types, demonstrating significant variability even within subtypes of tumors. The glioblastoma sample was predominantly comprised of a neural stem cell-like cell population. The Schwannoma sample was comprised of a specific stromal cell type.

Despite some overlap in the major cell types between tumor and non-tumor nuclei, their gene expression profiles were distinct. For example, the OPC4 cluster is unique to non-tumor nuclei, while tumor OPCs reside in OPC1-3. Some neuron-like clusters (i.e. NEU_EX3) that were present in tumors had very limited presence in the non-tumor samples. Our results suggest distinct tumor-associated gene expression alterations even if the tumor cell may resemble a normal brain cell type. However, comparisons with especially Schwannoma and pediatric-type high-grade glioma samples need to be further studied as although thousands of nuclei were included in the analysis, they were derived from only one sample each.

Our study supported some key findings from previous scRNA-seq experiments in ependymomas. Gojo et al. along with other studies identified radial glial-like cells as potential cells of origin in ependymomas[16,41,42]. Our results corroborate this finding with an abundance of radial glial cells in our ependymoma samples. Moreover,

**Table 3 | Number of significantly differentially expressed genes shared among all or subsets of tumor types**

| # of tumor types | # of genes shared among tumor types |
|---|---|
| 0 | 868 |
| 1 | 379 |
| 2 | 424 |
| 3 | 458 |
| 4 | 596 |
| 5 | 717 |
| 6 | 558 |

Gojo et al. indicate that stem-like cell populations are associated with more aggressive ependymomas[16]. Our results indicate a similar pattern in our expanded pediatric CNS tumor types, in which higher grade tumors are associated with cells with more stem-like features. Our study also supported results from Reitman et al., who demonstrated that pilocytic astrocytoma tumors are overall comprised of OPCs and mature glial-like cells[43]. Our results indicated a similar pattern in which much of our pilocytic astrocytoma samples were comprised of varying OPC clusters and couple of astrocyte-like clusters. The similarity of our results with previously published studies supports our results and previous findings in separate patient populations.

We identified the pathways enriched in varying cell types, with a focus on neural stem-like cells. Since NSCs have been shown to be associated with therapy resistance, metastasis, and tumor malignancy, it is important to specifically consider NSCs when treating pediatric CNS tumors and reducing risk for secondary neoplasms[44–50]. We determined potential targetable NSC-specific pathways. While some commonly enriched pathways like MYC and FOXM1 in NSCs may be considered very difficult to target as MYC and transcription factors are considered to be less druggable, there were more easily targetable pathways enriched in NSCs like Aurora-B kinase and retinoic acid pathway. However, the enriched pathways need to be validated further as the fold enrichment were incremental, although statistically significant.

With our cell type-adjusted approach, we addressed a critical confounder in differential gene expression analyses to identify transcriptomic alterations that exist in tumors compared to non-tumor tissue. While it may be intuitive that cell type composition differences introduce confounding effects in molecular comparisons between tumor and non-tumor tissue, very limited studies have empirically investigated the cell type composition effects, especially in pediatric CNS tumors, to our knowledge. Although the number of significantly differentially expressed genes decreased in the cell type-adjusted model compared to the cell type-unadjusted model, the adjusted model identified previously unlinked genes associated with tumors that would not have been uncovered in the unadjusted model. Moreover, the significantly differentially expressed genes exclusive to the unadjusted model likely stem from variations in cell type proportions, rather than from the underlying tumor biology that would be necessary for discovering effective therapeutic targets.

The pathways associated with the differentially expressed genes across the multiple tumor types in the cell type-adjusted model (translation-associated processes like peptide chain elongation and translation initiation/termination along with nonsense-mediated decay (NMD) processes) suggest the importance of these pathways commonly being dysregulated in pediatric central nervous system tumors. Previous studies have suggested the importance of down-regulation of NMD responses in the differentiation of neural stem cells[51–53]. Moreover, high levels of NMD factors were sufficient to keep the stemness of neural stem cells[51]. Interestingly, our results indicate upregulation of NMD associated genes across all pediatric CNS tumor types in comparison to non-tumor pediatric brain which suggest the

potential mechanism of upregulation of NMD maintaining more stem-like cells in these tumors. As more stem-like cells contributes to therapy resistance and recurrence, further studies investigating the NMD pathways and how they can be exploited to be potential therapeutic targets in pediatric CNS tumors are necessary.

Our study characterizes the heterogeneity that exists across pediatric CNS tumor types in comparison to non-tumoral pediatric brain tissue at the single-cell level. We also identify potential tumor type and cell type-specific molecular characteristics that may be used therapeutic targets for the various pediatric CNS tumors from primary tissue samples. Although there were very limited samples, our study included thousands of nuclei from these tumor types to gain a better understanding of cells that exist in these tumor types that previous studies have not studied deeply. From our results, complementary preclinical in vitro and in vivo experiments are needed to validate these targets to advance these potential targets as therapeutic options in the clinic.

## Methods

### Study population

This study complies with all Dartmouth Hitchcock Medical Center Institutional Review Board regulations. This study was approved by the Dartmouth Hitchcock Medical Center Institutional Review Board Study #00030211. All subjects provided consent for the use of tissues for research purposes. Tumor and non-tumor tissues were collected from patients treated at Dartmouth Hitchcock Medical Center from 1993 to 2017. Parents/legal guardians of the patients consented to use of tissues for research purposes. Histopathologic tumor type and grade for each sample were re-reviewed according to the 2021 WHO classification of CNS tumors and categorized into the major tumor types[5]. Tumor types included in this study are astrocytoma, embryonal tumors, ependymoma, glioneuronal/neuronal tumors, glioblastoma, and Schwannoma. The average age at diagnosis of subjects from whom the tumor tissues were derived from in this study was 9.3 (range: 0.75–18). Male subjects accounted for 62.9% of the tumor samples and female subjects accounted for 37.1% of the tumor samples. Non-tumor brain tissues were obtained from pediatric patients with epilepsy who underwent surgical resection. The average age at diagnosis of subjects from whom the non-tumor samples were derived from was 6.2 (0.58–11). Male subjects accounted for 33.3% of the non-tumor samples and female subjects accounted for 66.7% of the non-tumor samples. Specific demographic characteristics of patients for the study are provided in Table 1 and sample information for each subject are provided in Supplementary Table 1.

### Identification of genetic variation with bulk tissue RNA-seq

RNA was collected using Qiagen RNeasy plus kit (Catalog ID: 74034, Qiagen, Hilden, Germany). RNA-seq libraries were prepared following the Takara Pico v3 low input protocol and sequenced on Illumina NextSeq500.

Raw RNA-seq data were trimmed for polyA sequences and low-quality bases using *cutadapt* (v2.4)[54]. Reads were aligned to human genome hg38 using STAR (v 2.7.2b)[55]. Duplicate read identification and other quality control checks for read alignment were performed using CollectRNASeqMetrics and MarkDuplicates in *Picard Tools*.[56] Reads containing N were split using SplitNCigarReads function in the Genome Analysis Toolkit (GATK)[57,58]. Bases quality scores were recalibrated using known variants from the GATK resource bundle and with the BaseRecalibrator and ApplyBQSR functions in GATK[57,58]. Somatic SNV and indels were called with Mutect2 in tumor-only mode[57,58]. Only variants with at least read depth of 10, 5% allele frequency, read depth of 5 for the alternate allele were kept for analysis. Somatic variants determined to be false positives were removed with the GATK Filter-MutectCalls. The variants were then filtered for variants in sex or mitochondrial chromosomes, RNA editing sites, repeat masker

regions, and variants in Panel of Normal (from GATK) references. Variants were then annotated using the Funcotator function in GATK[57,58].

## Identification of copy number variation with DNA methylation arrays

DNA were treated with sodium bisulfite following the TrueMethyl® oxBS Module (Tecan Genomics Inc, Redwood City, CA). Converted DNA were hybridized to Infinium HumanMethylationEPIC BeadChips. Raw idat files from the EPIC arrays were processed using pre-processNoob function in minfi in R[59]. Copy number variations of tumor samples were estimated in comparison to non-tumor samples using the CNV.fit function in conumee package in R[60].

## Nuclei isolation, sample multiplexing, and single nuclei-RNA-sequencing

Nuclei from fresh frozen tissues were isolated following the Nuclei Pure Prep nuclei isolation kit (Sigma-Aldrich, Catalog ID: NUC201) with some modifications. To summarize, ~10 mg of tissue were washed with PBS to remove extraneous OCT the samples were frozen in. The tissue was homogenized with both wide and narrow pestles submerged in 2.5 mL of the lysis buffer in a Dounce homogenizer. The lysate mixed with 4.5 mL 1.8 M sucrose cushion were gently layered on top of the 2.5 mL of 1.8 M sucrose cushion in Beckman ultracentrifuge tubes. Samples were centrifuged for 45 min at 22,673 × $g$ at 4 °C in an ultracentrifuge. Samples were multiplexed with lipid-tagged oligonucleotides following the MULTI-seq protocol (Supplementary Table 3)[20]. Nuclei were resuspended in 1% BSA PBS and filtered with 70 μm and 40 μm Flowmi filters. Nuclei were quantified with Cellometer K2 (Nexcelom, Lawrence, MA). We aimed for 2500–5000 nuclei per sample to be sequenced. Four tumor samples were not multiplexed and were separately prepared (Supplementary Table 1).

Libraries for single nuclei RNA-seq were prepared following the 10x Genomics Single Cell Gene Expression workflows (10x Genomics, Pleasanton, CA) and were sequenced on Illumina NextSeq500 to average 45,000 reads per cell. 10X Cell Ranger software was used to align sequences to GRch38 pre-mRNA reference genome and generate feature-barcode matrices for downstream analyses.

## Pre-processing snRNA-seq data

To filter low-quality nuclei, only those with greater than 200 and less than 10,000 features and less than 5% of reads that map to the mitochondrial genes were used in downstream analyses. Pooled nuclei were demultiplexed by hashtag oligonucleotides using HTODemux function in Seurat v4[61–64]. Pooled samples were also demultiplexed using Vireo, a genotype-based demultiplexing method[65]. We performed genetic demultiplexing analysis using genotype data following the methods described in Weber et al.[66], implemented in a Nextflow workflow[67]. Briefly, bulk RNA-seq reads from each sample were mapped to the reference genome (GRCh38.p13) using STAR[55]. Pooled single-nuclei RNA-seq reads were mapped to the reference genome using STARsolo[68]. Variants among the samples within each pool were identified and genotyped with bcftools mpileup[69] using the mapped bulk reads. Individual cells were then genotyped only at the sites identified using the bulk RNA using cellsnp-lite (mode 1a)[70]. Cell genotypes were used to identify the sample of origin for each cell using Vireo[65]. Code for the genetic demultiplexing workflow can be found at https://github.com/AlexsLemonade/alsf-scpca/tree/main/workflows/genetic-demux.

To integrate the methods, we first used sample identity assigned from the hashtag oligonucleotides. If the nuclei were confidently assigned a sample, it was compared to the genotype-based sample assignment. Those that did not match the same sample were filtered out. If the nuclei were assigned as a doublet or to none of the samples, the nuclei were assigned to a sample based on the genotype-based approach. 84,700 nuclei with confident sample assignment were used in analysis.

As our dataset included a very large number of nuclei to be integrated and was expected to have certain cell types only present in certain samples, we used the reciprocal PCA integration approach on the 2000 most variable features to combine the nuclei from each sample. We first found the integration anchors with the FindIntegrationAnchors function then used the IntegrateData function in Seurat v4 to integrate all our filtered nuclei[62–64].

## Dimension reduction and clustering of snRNA-seq data

The integrated dataset was scaled using the ScaleData function in Seurat. First, PCA dimensionality reduction analyses were done to identify 100 principal components (PCs). To further reduce the dimensionality and cluster our nuclei by their gene expression profile, we conducted UMAP analyses on the 50 PCs with highest standard deviation with RunUMAP function in Seurat[61,71]. Then, we clustered our cells using FindNeighbors (n_neighbors = 30) and FindClusters (resolution = 1.0) function in Seurat[61].

## Gene set enrichment testing

Gene set enrichment tests at the single-cell level were conducted using the Variance-Adjusted Mahalanobis (VAM) method[21]. The vamForSeurat function from the VAM R package was used to calculate enrichment scores for each nucleus. Brain cell type-specific gene sets from the Molecular Signatures Database (MSigDB) v7.5.1 were used to validate our single-cell identities[23–28]. For identifying cell types, $p$ values were calculated from the cumulative distribution function values generated by VAM. Nuclei were considered to be associated with a specific brain cell type-pathway if the VAM-generated $p$ value was ≤ 0.05. Nucleus-level pathway scoring was also conducted using VAM for pathways in the MsigDB Pathways Interaction Database (PID) collection[34]. PID Pathways were considered to be enriched in each nucleus at the FDR adjusted p-value threshold of 0.1 for the VAM-generated $p$ values.

Stemness scores for each nucleus were calculated using the stemness-associated gene list from Tirosh et al.[32] and the AddModuleScores function in Seurat.

## Differential gene expression and pathways

Differential expression analysis between tumor nuclei and non-tumor nuclei were conducted using monocle3[72–75]. Differential expression analyses were conducted only on the top 4000 most variable features identified from the FindVariableFeatures Seurat function. The unadjusted differential expression testing was done using the fit_models function in monocle3 (v1.0.0) R package with the quasi-poisson distribution with the non-tumor nuclei being the referent gene expression profile[72,74,75]. The adjusted differential expression testing was done with the same quasi-poisson distribution with non-tumor nuclei being the referent but including the major cell type identity in the model. Gene types for each gene used in the differential expression testing were annotated using the org.Hs.eg.db[76], Human genome annotation package, and mapIds function in the AnnotationDbi R package[77]. Pathways associated with the differentially expressed genes were identified using the Reactome pathways and ReactomePA R package[40].

Pathways important for each cell cluster were identified using FindAllMarkers function in Seurat with the Wilcoxon rank sum test in Seurat on the binary classification of PID pathways enrichment for each nuclei[78]. Log fold change and minimum percentage of cells enriched in each pathway were both set to 0. To identify the pathways with greater number of nuclei with enriched pathway per cluster, we selected pathways that were only positive in direction in the FindAllMarkers options.

## Statistical testing

Observed proportion of genes that were either increased or decreased in the same direction in the shared differentially expressed genes among all the tumor types (60.9%) were compared to expected proportion of genes that would be increased or decreased in the same direction across all the tumor types (3.13%) using a one-sample proportion test. The expected proportions were determined based on the permutations of direction of change compared to nontumor for the six tumor types.

## Comparison between our cohort and publicly available datasets

We utilized the cell types identified in our single nuclei RNA-seq data as reference to annotate cell types of publicly available single-cell RNA-seq datasets in GSE141460 (ependymoma)[16] and GSE214357 (medulloblastoma)[29]. Both GEO datasets were pre-processed and integrated in the same manner as our dataset as discussed above. We identified transfer anchors (pairs of cells from each dataset within each mutual nearest neighbors) between the publicly available datasets and our dataset by using the FindTransferAnchors function in Seurat. We then classified the cells in the publicly available datasets based on cell types labeled in our pediatric CNS tumor dataset by using the TransferData function in Seurat. The distribution of cell types between the publicly available datasets and respective tumor type in our cohort were compared using the Wilcoxon signed rank test. The statistical significance threshold for these comparisons were set to $p$ value < 0.05.

## Reporting summary

Further information on research design is available in the Nature Portfolio Reporting Summary linked to this article.

## Data availability

The raw single nuclei-RNA-seq data and the processed data for single nuclei-RNA-seq generated in this study are available in the Gene Expression Omnibus under accession code GSE211362. The raw hydroxymethylation/methylation data generated in this study have been deposited in the Gene Expression Omnibus under accession code GSE152561. The raw bulk RNA-seq data generated in this study have been deposited in the Gene Expression Omnibus under accession code GSE241396. The processed single nuclei RNA-seq data is also available through the Pediatric Single Cell Atlas provided by the Alex's Lemonade Stand Foundation (https://scpca.alexslemonade.org). Detailed annotation of which pool each sample was multiplexed in can be found in Supplementary Table 1. All other data generated in this study are provided in the Supplementary Information/Source Data file. The source data large in size are available in Figshare (https://figshare.com/projects/Associations_in_cell_type-specific_hydroxymethylation_and_transcriptional_alterations_of_pediatric_central_nervous_system_tumors/193781). Source data are provided with this paper.

## Code availability

Code used for this study is available at https://github.com/sarahmklee/IntegrativePCNS.

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

## Acknowledgements

We would like to thank Dr. Matthew Havrda (Department of Molecular and Systems Biology, Geisel School of Medicine) for assistance in nuclei isolation protocol development. We would like to thank Ally Hawkins (Childhood Cancer Data Lab, Alex's Lemonade Stand Foundation) for code review and Jaclyn Taroni (Childhood Cancer Data Lab, Alex's Lemonade Stand Foundation) for assistance in data analysis. This work was supported by a Prouty Pilot award from the Dartmouth Cancer Center and a Single-cell Pediatric Cancer Atlas (ScPCA) grant from the Alex's Lemonade Stand Foundation. MKL was supported by the Burroughs-Wellcome Fund: Big Data in the Life Sciences at Dartmouth. NA was supported by the S.M. Tenney Fellowship at Dartmouth. This work was also supported by R01CA216265, R01CA253976, and P20GM104416 – 6369 to B.C.C. and P20GM130454 – 7702 and R21CA253408 to H.R.F. Single-nuclei RNA-seq experiments were conducted in the Genomics and Molecular Biology Shared Resource (GMBSR) at Dartmouth, which is supported by NCI Cancer Center Support Grant 5P30CA023108 and NIH S10 (1S10OD030242) awards. Single-nuclei-RNA-experiments were also supported through the Dartmouth Center for Quantitative in collaboration with the GMBSR with support from NIGMS (P20GM130454) and NIH S10 (S10OD025235) awards.

## Author contributions

M.K.L., N.A. and F.W.K. carried out the experiments. N.A., L.N.N., G.J.Z. obtained samples and clinical data. MKL performed data analyses with the help of H.R.F. and B.C.C. J.A.S. processed single-cell-level genotypes demultiplexing. B.C.C. supervised the projects. All authors read and approved the final manuscript.

## Competing interests

B.C.C. is an advisor to Guardant Health which had no role in this work. The remaining authors declare no competing interests.
