## [Peer Review File · Nature Communications]

REVIEWER COMMENTS

Reviewer #1 (Remarks to the Author): Expert in paediatric CNS tumour genomics

This is a study conducted by Drs. Lee et al on the tumor type and cell-type specific gene expression alterations in a set of 35 pediatric brain tumor using nuclei RNA-seq. As the authors correctly pointed out, there is a need of better understanding of tumor biology for this group of brain cancers that are still difficult to treat. While single cell transcriptomics study is still at its infancy, this paper extended previous studies by adding additional cases of the common brain tumor cancers and also by including some rare tumors (desembryoplastic neuroepithelial tumors and gangliogliomas). The team should also be applauded by their inclusion of normal brain tissues as reference. Although the overall strategy is not new, some new findings were presented, including the identification of ~200 genes per tumor that were not observed in unadjusted models. Their attempt of pooling samples via lipid-tagged hastag oligonucleotides (HTO) appeared to be a very efficient strategy that can be cost-effective as well. The data analysis approaches were sound and logic. There are several concerns:

Major concern:

1. The small sample size for each and every tumor subtype that were examined in the current study limited the impact on pediatric brain tumors. Although they team attempted to group all tumors together and dissecting each tumor and cell type separately, there is not enough power of analysis to draw conclusions for different type of brain tumors.
2. Although comparing among the tumors revealed some cell-type differences, this information did not add much novelty as the cellular differences among pediatric brain tumors have a long been recognized.

Minor concerns:

1. Non-tumor tissues are ideally matched with the age of the donor and location of tissue origin. Table 1 was missing, and no detailed information can be found.
2. How were the tumor stemness score calculated?
3. It is over ambitious to draw any meaningful conclusions about tumor-type specific changes by comparing at least 5 different types of pediatric brain tumors with a mere 32 samples.
4. Insufficient understanding/reference of established genetic changes were presented.
5. The authors reported 84,700 nuclei with confident sample assignment. What was total? Or, how many nuclei were not properly assigned?

Reviewer #2 (Remarks to the Author): Expert in single-cell omics and brain cancer genomics

Pediatric central nervous system (CNS) tumors are the leading cause of cancer-related deaths in children and carry an increased risk of secondary tumors. However, progress in targeted therapies for these tumors has been slower compared to adult tumors, mainly due to their low prevalence. In this study, the authors collected and analyzed single nuclei RNA-seq data from 35 pediatric CNS tumors and three non-tumoral pediatric brain tissues. By studying the transcriptomic alterations and tumor heterogeneity, they

identified specific cell subpopulations associated with different tumor types, such as radial glial cells in ependymomas and oligodendrocyte precursor cells in astrocytomas. They also observed pathways related to neural stem cell-like populations, known for their resistance to therapy. Furthermore, they compared the gene expression profiles of pediatric CNS tumor types with non-tumor tissues, enhancing understanding of gene expression profiles in single cells of various pediatric CNS tumors.

The following points should be clarified and edited in the study:

1. From a biological perspective, this reviewer does not find the results reported in this paper to be particularly exciting or novel. As noted by the authors, their results mostly provide a confirmation of multiple prior observations. While there are some novel findings, this reviewer finds them to be either incremental or for the authors to be a bit pushing their results.

The investigators provide a novel and exciting approach to simultaneously assess multiple transcriptional alterations at the single-cell level. However, as the authors demonstrate, the percentage of cells that can be successfully demultiplexed is highly variable.

Is there any way to further troubleshoot the protocols used? How efficient is this protocol? Further, the interpretation of some of the findings, particularly given the minimal fold changes (e.g., Figure 5) must be further addressed and edited.

2. Some of the statements and results are not adequately supported by quantitative/statistical measurements. Several of the findings must be interpreted with greater precision. Overall, the paper would benefit from a more detailed explanation of the rationale and scope of the analyses, a stronger and more detailed description of the methodology, and the addition of specifics to the legends and figure panels.

3. Figure 1. The authors reported results of mutation calling analysis from bulk RNA-seq data. These bulk RNAseq data should be made available and uploaded to relevant database in both raw and processed format.

The authors should comment on the accuracy of calling mutations from bulk RNAseq data. What is the false positive/negative rate?

It would be also beneficial to provide more details on the genetic variants identified. A simple table with mutation information (VAF, amino acid change, etc...) would be a useful resource for the community.

Overall, the relevance of this analysis is unclear. The authors should better clarify the methods, approaches, rationale, scope and better describe the relevance and meaning of the results obtained.

4. The text involving Figure 1 would benefit from details. It is unclear what the authors mean with many of the variants being associated with epigenetic processes. What epigenetic processes? And how is this association made or tested?

5. There is some discrepancy between sample assignments from HTO and genotyping (Figure 2B) that may become an issue with downstream analysis. Fig. 2B GT and HTO did not align very well with a large portion of doublet (from HTO) and unassigned. Need to be cautious with the confidence of the downstream analysis.

6. The message behind analysis of stemness score is unclear. The authors state “Levels of expression for genes classically used for to isolate stemlike cells in literature varied among the different cell types”. Not sure what is the message that the authors want to deliver in this sentence. A lot of genes failed to be detected in Supplementary Figure 2.

7. In the analysis of Supplementary Figure 4C, should the effects from sex, location, tumor type and grade be accounted and adjusted for?

8. “Because the embryonal tumor cells (EMB) clusters were unlike any other classical cell type found in the brain, the cells in these clusters were classified as embryonal tumor cells”. Is this saying EMB cells were classified as EMB cells because the clusters were unlike other cell types? From Fig. 3B, PAX6 is the only marker expressed in this cluster? In the Supplementary Figure 1B, EMB signals are also not very strong.

9. The authors should specify how the “sequencing-derived genotype data from each nucleus” were obtained.

10. For the cell-type specific pathways analysis, how relevant is to test pathways in human using mouse experiments?

11. This reviewer is a little suspicious of the stem-like cells not expressing stem-like markers (SOX2 and CD44). Is it biologically relevant or an artifact due to sample quality, demultiplexing or analysis?

12. Figure 3 panel C is missing from Figure 3.

13. Overall, the authors need to be cautious to pull samples from different tumor types and grades for integrated analysis. Given there is only 1 GBM sample and 1 Schwannoma sample, some results of differential analysis need to be carefully evaluated. How is the difference in the number of nuclei controlled in the integrated analysis?

14. The analytical codes used to generate the main results of the paper must be provided.

REVIEWER COMMENTS

Reviewer #1 (Remarks to the Author): Expert in paediatric CNS tumour genomics

This is a study conducted by Drs. Lee et al on the tumor type and cell-type specific gene expression alterations in a set of 35 pediatric brain tumor using nuclei RNA-seq. As the authors correctly pointed out, there is a need of better understanding of tumor biology for this group of brain cancers that are still difficult to treat. While single cell transcriptomics study is still at its infancy, this paper extended previous studies by adding additional cases of the common brain tumor cancers and also by including some rare tumors (desembryoplastic neuroepithelial tumors and gangliogliomas). The team should also be applauded by their inclusion of normal brain tissues as reference. Although the overall strategy is not new, some new findings were presented, including the identification of ~200 genes per tumor that were not observed in unadjusted models. Their attempt of pooling samples via lipid-tagged hastag oligonucleotides (HTO) appeared to be a very efficient strategy that can be cost-effective as well. The data analysis approaches were sound and logic. There are several concerns:

Major concern:

1. The small sample size for each and every tumor subtype that were examined in the current study limited the impact on pediatric brain tumors. Although they team attempted to group all tumors together and dissecting each tumor and cell type separately, there is not enough power of analysis to draw conclusions for different type of brain tumors.

We agree with the reviewer that the sample size is relatively small compared with analyses of other tumor types and is a limitation that we and others studying pediatric brain tumors contend with. However, for each tumor studied we include molecular information from hundreds to thousands of cells which addresses some potential concern with distinctions among subtypes. It is quite difficult to accrue a larger sample size for these tumor types. It is also very difficult to obtain non-tumor pediatric brain tissues to be used in research. Our sample size may be limited; however, our findings offer a starting point to begin to investigate the effects of heterogeneity so that larger studies in collaboration with other medical centers or consortiums can be conducted to increase statistical power. For additional context on our sample size, in comparison to the few published studies that have investigated single cell states of pCNS tumors, here, we contribute the largest sample size (>40% increase in total pCNS samples available for analysis), and one of the largest number of nuclei analyzed (84,700 nuclei, a >74% increase in total pCNS nuclei available for analysis). For instance, Filbin et al analyzed ~2,500 cells in their study investigating H3K27M pediatric gliomas (Science, 2018), Hovestadt et al characterized ~9,900 cells from medulloblastomas (Nature, 2019), and Gillen et al assessed 18,500 cells from ependymomas (Cell Reports, 2020). We believe that our dataset enhances the limited literature that has been published on these rare and difficult to treat tumors.

We have added the following to the text:

(pg. 14): Due to the rarity of pediatric CNS tumors, it is difficult to accrue a large sample size for a highly powered study. Although our sample size was limited, our study incorporates pediatric CNS tumor types that have not yet been characterized with single cell or single nuclei RNA-seq such as gangliogliomas and further enhances the current limited literature on some pediatric CNS tumor types.

2. Although comparing among the tumors revealed some cell-type differences, this information did not add much novelty as the cellular differences among pediatric brain tumors have a long been recognized.

There has been in depth research into the cellular composition for some specific types of pediatric central nervous system tumors. However, previous studies have focused on cellular composition by using traditional methods like FACS. These methods are limited due to the bias that is posed from utilizing only the known stainable markers for specific cell types. Moreover, some cell types of the brain such as neurons are more difficult to isolate with FACS due to the structure of the cell. With a single nuclei RNA-seq approach, we took an unbiased approach to describe the cellular landscape of pediatric CNS tumors. In addition, as our study utilizes nuclei instead of cells, we are able to incorporate cell types that would be difficult to isolate with FACS. Furthermore, our study includes tumor types that are much under-investigated in terms of molecular characteristics (GNNs). Lastly, none of the previous studies investigating pediatric central nervous system tumors with single cell genomics have included non-tumor pediatric tissues for reference. One of the major objectives here was to investigate alterations in comparison to non-tumor tissue, not relative to other tumor types.

Minor concerns:

1. Non-tumor tissues are ideally matched with the age of the donor and location of tissue origin. Table 1 was missing, and no detailed information can be found.

While it is ideal to match non-tumor tissue to the location of the tumors, it was difficult to do so with limited samples. Table 1 describes the age, sex, location, and tumor type and is referenced in the first paragraph of the results section (pg. 4) and in the methods section (pg. 17). Detailed sample specific information, including tumor type, grade, # of nuclei per sample, is provided in Supplementary Table 1A.

2. How were the tumor stemness score calculated?

Tumor stemness scores were calculated using genes previously identified to be associated with stemness (i.e.: *SOX2/11* and *CCND2*) (Tirosh et al, 2016; PMID: 27806376). We used the *AddModuleScore* function in Seurat to calculate the stemness score. This function calculates the module score by subtracting the aggregated expression of control feature sets from the average expression level of a set of genes (Tirosh et al 2016; PMID: 27806376).

3. It is over ambitious to draw any meaningful conclusions about tumor-type specific changes by comparing at least 5 different types of pediatric brain tumors with a mere 32 samples.

While we agree with the reviewer that we do not have the biggest sample size, the unit of analysis is nuclei, and 84,700 nuclei were included, presenting a large set of gene expression for single nuclei among very rare tumor types that are hard to accrue samples for. As mentioned above, we have added >40% increase in total pCNS samples available for analysis and >74% increase in total pCNS nuclei available for analysis. While the number of tumors is limited, the number of nuclei analyzed is large and will

provide utility for investigators who all are constrained by limited tumor numbers and size for rare pediatric central nervous system tumors.

4. Insufficient understanding/reference of established genetic changes were presented.

Thank you for pointing this out. We incorporated only limited information of the genetic changes in these samples as it was not the focus for this study. We did not conduct whole genome sequencing as we had only limited sample substrate to work with for each sample, which is not uncommon for these rare tumors. Instead, we utilized reads from bulk RNA-seq to obtain genetic alteration data. We have included an additional table (Supplementary Table 1B) with more detail on the genetic alterations in each sample. Following the reviewer's comments, we have moved Figure 1 (previously the somatic variant information) to be Supplementary Figure 1 to not distract from the main objective of the study.

5. The authors reported 84,700 nuclei with confident sample assignment. What was total? Or, how many nuclei were not properly assigned?

We sequenced a total of 151,272 nuclei and were able to confidently assign 84,700 – or 56% of the total nuclei. We have added the following to the manuscript:

(pg. 14): We included de-multiplexed nuclei in alignment by both methods as well as de-multiplexed nuclei by either method if the other method had an undetermined assignment to increase the number of nuclei for this analysis. As some nuclei were not in alignment for both, limitations in accuracy of sample assignment remain in our downstream analysis.

Reviewer #2 (Remarks to the Author): Expert in single-cell omics and brain cancer genomics

Pediatric central nervous system (CNS) tumors are the leading cause of cancer-related deaths in children and carry an increased risk of secondary tumors. However, progress in targeted therapies for these tumors has been slower compared to adult tumors, mainly due to their low prevalence. In this study, the authors collected and analyzed single nuclei RNA-seq data from 35 pediatric CNS tumors and three non-tumoral pediatric brain tissues. By studying the transcriptomic alterations and tumor heterogeneity, they identified specific cell subpopulations associated with different tumor types, such as radial glial cells in ependymomas and oligodendrocyte precursor cells in astrocytomas. They also observed pathways related to neural stem cell-like populations, known for their resistance to therapy. Furthermore, they compared the gene expression profiles of pediatric CNS tumor types with non-tumor tissues, enhancing understanding of gene expression profiles in single cells of various pediatric CNS tumors.

The following points should be clarified and edited in the study:

1. From a biological perspective, this reviewer does not find the results reported in this paper to be particularly exciting or novel. As noted by the authors, their results mostly provide a confirmation of multiple prior observations. While there are some novel findings, this reviewer finds them to be either incremental or for the authors to be a bit pushing their results. The investigators provide a novel and exciting approach to simultaneously assess multiple transcriptional alterations at the single-cell level. However, as the authors demonstrate, the percentage of cells that can be successfully demultiplexed is highly variable. Is there any way to further troubleshoot the protocols used? How efficient is this protocol?

Further, the interpretation of some of the findings, particularly given the minimal fold changes (e.g., Figure 5) must be further addressed and edited.

We thank the reviewer for this feedback. While some results from our study confirms some of the findings from previous studies, we believe that our study adds additional data for tumor types that have been very under-investigated. To our knowledge, there are only 1 – 2 publications for some tumor types like ependymoma and medulloblastoma utilizing single cell RNA-seq to characterize cellular heterogeneity. In addition, we contribute the largest sample size (>40% increase in total pCNS samples available for analysis), and one of the largest number of nuclei analyzed (84,700 nuclei, >74% increase in total pCNS nuclei available for analysis), data we of course make available for others in the field. We also include tumor types that currently do not have any single cell RNA-seq studies like glioneuronal/neuronal tumors. Furthermore, to our knowledge, no existing studies have incorporated non-tumor pediatric brain tissues for reference. Our results provide a direct comparison of the transcriptome in pediatric CNS tumors to non-tumor pediatric brain tissue which has not yet been shown.

Importantly, while it may be intuitive that cell type composition introduces confounding effects when comparing molecular features of tumors to non-tumor tissue, no other previous studies have empirically shown this, especially in pediatric CNS tumors. We have added this significance in the discussion (pg.15):

‘While it may be intuitive that cell type composition differences introduce confounding effects in molecular comparisons between tumor and non-tumor tissue, no other studies have empirically investigated the cell type composition effects, especially in pediatric CNS tumors, to our knowledge.’

In regard to the de-multiplexing approach, we attempted many different troubleshooting methods to improve the methods. However, different approaches did not significantly improve the recovery of quality nuclei. It is highly likely that we reached a threshold for the quality nuclei we could recover.

We have added limitations in the manuscript:

(pg. 14): Due to the rarity of pediatric CNS tumors, it is difficult to accrue a large sample size for a highly powered study. Although our sample size was limited, our study incorporates pediatric CNS tumor types that have not yet been characterized with single cell or single nuclei RNA-seq such as gangliogliomas and further enhances the current limited literature on some pediatric CNS tumor types.

(pg.15): ‘However, the enriched pathways need to be validated further as the fold enrichment were incremental, although statistically significant.’

2. Some of the statements and results are not adequately supported by quantitative/statistical measurements. Several of the findings must be interpreted with greater precision. Overall, the paper would benefit from a more detailed explanation of the rationale and scope of the analyses, a stronger and more detailed description of the methodology, and the addition of specifics to the legends and figure panels.

We thank the reviewer for the suggestion. We have revised the manuscript text and figure legends to include more details especially in response to reviewers' feedback. Following are some locations of where revisions were made:

(pg. 4): Tumors are known to be composed of heterogeneous cell types¹⁸⁻²⁰. Yet many times, cell type heterogeneity is not considered when molecular profiles of bulk tumor tissue are compared to bulk non-tumor tissue. Thus, we here investigate the effects from cell type composition differences when comparing the transcriptome of pediatric CNS tumors and non-tumor pediatric brain tissue by integrating single nuclei RNA-seq and bulk tissue RNA-seq datasets.

(pg. 4): Additional molecular information of our pediatric CNS tumor cohort was determined from the bulk RNA-seq and DNA methylation data.

(pg. 6): We utilized uniform manifold approximation and projection (UMAP), a dimension reduction strategy, to identify nuclei most similar to each other based on gene expression levels.

(pg. 7): As cellular stemness is a key characteristic in sustaining tumor survival and malignancy^{32,33}, we set out to investigate stemness in each cell types of our pediatric CNS tumors.

(pg. 9): We set out to investigate potential therapeutically targetable pathways specific to cell types associated with tumorigenesis and progression in the pediatric CNS tumors.

(pg. 10): Next, we determined any pathways that were more specific for each cell type by comparing the enrichment classifications from above for each cell type to determine important cell type-specific pathways.

3. Figure 1. The authors reported results of mutation calling analysis from bulk RNA-seq data. These bulk RNAseq data should be made available and uploaded to relevant database in both raw and processed format.

The authors should comment on the accuracy of calling mutations from bulk RNAseq data. What is the false positive/negative rate?

It would be also beneficial to provide more details on the genetic variants identified. A simple table with mutation information (VAF, amino acid change, etc...) would be a useful resource for the community.

Overall, the relevance of this analysis is unclear. The authors should better clarify the methods, approaches, rationale, scope and better describe the relevance and meaning of the results obtained.

Thank you for the suggestion. We have included the variant details (type of variant, cDNA/codon/protein change) identified from our bulk RNA-seq data in Supplementary Table 1B. Following the reviewer's suggestions, we have also change Figure 1 to include variant type. We have added the data for bulk RNA-seq in GSE241396 (reviewer token: wfuxikoabhwxfut).

Because we utilized RNA-seq data to call variants, we utilized more stringent thresholds to determine whether it was a variant or not. Since we used nuclear RNA (instead of mRNA), we were able to capture more than just exonic variants that mRNA-seq would

identify. Only variants with at least read depth of 10, 5% allele frequency, read depth of 5 for the alternate allele were used in downstream analysis. Variants determined to be false positives determined by the GATK FilterMutectCalls were removed. We agree that there are limitations for calling variants from bulk RNA-seq data, especially without matching referent, rather than from whole genome or exome sequencing data. We have added following limitation in the manuscript as follows:

(pg. 5): While the variants were filtered under more stringent cut-offs, there may have been undetected variants or misclassified variants as they were called from bulk RNA-seq in comparison to a reference.

(pg. 18): Somatic variants determined to be false positives were removed with the GATK FilterMutectCalls.

The major focus of this study was on the cellular heterogeneity and the transcriptional changes in the pediatric CNS tumors in comparison to the non-tumor pediatric brain tissue while taking the cell type composition into account. The genetic changes were included in the manuscript as additional information on these pediatric CNS tumors as they may be of interest and of utility in the future when datasets might be combined. We agree from the reviewer comments that it may be a little confusing as to why it was included. We have moved this figure to be a supplementary figure (Supplementary Figure 1) to not detract from the overall objective of the manuscript.

4. The text involving Figure 1 would benefit from details. It is unclear what the authors mean with many of the variants being associated with epigenetic processes. What epigenetic processes? And how is this association made or tested?

Here we are referring to genetic alterations in epigenetic regulatory genes. For example, *HIST1H1E* is a histone linker gene, and *MALAT1* is a non-coding RNA with roles in nuclear organization and modulating gene expression that have been identified. We have revised to clarify the text in the manuscript as follows:

(pg. 5): 'Many of the somatic genetic variants (insertions, SNPs, deletions, etc.) detected within the pediatric CNS tumors were identified in genes known to have functions in epigenetic processes. For example, almost half of the tumors (14/33), across tumor types, had genetic variants in HIST1H1E, a H1.4 linker histone gene. Interestingly, across all but one tumor sample, tumors had genetic variants in MALAT1, a non-coding RNA with roles in nuclear organization and modulation of gene expression.'

5. There is some discrepancy between sample assignments from HTO and genotyping (Figure 2B) that may become an issue with downstream analysis. Fig. 2B GT and HTO did not align very well with a large portion of doublet (from HTO) and unassigned. Need to be cautious with the confidence of the downstream analysis.

Thank you for pointing this out. We agree that there was some discrepancy between sample assignments from HTO and genotyping. We removed any that were fully not in agreement with each other. However, to increase the number of nuclei we included in the analysis, we decided to rely on either method if one method had described it as unknown or doublet. We have revised the text to include limitations from this as follows:

(pg. 14): We included de-multiplexed nuclei in alignment by both methods as well as de-multiplexed nuclei by either method if the other method had an undetermined assignment to increase the number of nuclei for this analysis. As some nuclei were not in alignment for both, limitations in accuracy of sample assignment remain in our downstream analysis.

6. The message behind analysis of stemness score is unclear. The authors state “Levels of expression for genes classically used for to isolate stemlike cells in literature varied among the different cell types”. Not sure what is the message that the authors want to deliver in this sentence. A lot of genes failed to be detected in Supplementary Figure 2.

Thank you for pointing this out. Yes, we agree that many of the genes failed to be detected in Supplementary Figure 2. However, for those that were expressed, the levels were variable among different cell types. Following are some revisions made to the text:

(pg. 7): Although expression in some stemness marker genes were not detected, in those that were detected, levels of expression for genes classically used for to isolate stem-like cells in literature varied among the different cell types (**Supplementary Figure 3, Supplementary Table 2**). Interestingly, cell types expected to be more differentiated, like astrocytes, had relatively high levels of CD44 and VIM, and these genes were expressed in many of the cell types. As there were some genes that failed to be detected in a lot of the cell types, we calculated stemness scores based on a larger set of stemness related genes.

7. In the analysis of Supplementary Figure 4C, should the effects from sex, location, tumor type and grade be accounted and adjusted for?

Thank you for the suggestion. We have included Supplementary Figure 4D, to account for tumor type as not to include too many variables within our sample size. When investigated by tumor type, we found the correlation coefficients to all be positive, although only in astrocytoma was that correlation statistically significant at P-value < 0.05. It is likely that with additional sample size for each tumor type, these correlations become statistically significant. We have added these results in the manuscript as follows:

(pg. 8): When this association was investigated by each tumor type, only astrocytoma was found to be statistically significant ($R = 0.72$, $P\text{-value} = 0.043$, **Supplementary Figure 5D**). However, as positive correlation coefficients were found for the other tumor types, an increased sample size may likely identify a statistically significant positive association.

8. “Because the embryonal tumor cells (EMB) clusters were unlike any other classical cell type found in the brain, the cells in these clusters were classified as embryonal tumor cells”. Is this saying EMB cells were classified as EMB cells because the clusters were unlike other cell types? From Fig. 3B, PAX6 is the only marker expressed in this cluster? In the Supplementary Figure 1B, EMB signals are also not very strong.

Nuclei in the EMB clusters (EMB1, EMB2) were found to be specific to embryonal tumors and clustered distinctly from other cell types of the other tumor types. Based on current

information on markers for brain cell types we could not confidently call cell types that the nuclei embryonal tumors were most similar to. Consequently, we named them separately as EMB because they were all from the same tumor type and clustered together. The text has been revised as follows for additional clarity:

(pg. 7): Nuclei in the EMB clusters (EMB1, EMB2) did not have detectable levels of currently known markers for classical brain cell types. These nuclei were found to be specific to embryonal tumors and clustered distinctly from other tumor types; therefore, were named the EMB (embryonal) cluster.

9. The authors should specify how the “sequencing-derived genotype data from each nucleus” were obtained.

Thank you for pointing this out. ‘Sequencing-derived genotype’ data refers to the actual genetic sequences from the bulk RNA-seq analysis. We have revised it to indicate the bulk RNA-seq data in pg. 5 of the text.

10. For the cell-type specific pathways analysis, how relevant is to test pathways in human using mouse experiments?

We thank the reviewer for the suggestion. We do agree that testing these pathways in mouse experiments will be important to validate the importance of specific pathways for each pediatric CNS tumor types. It will be difficult to test across all tumor types as quality mouse models for some pediatric CNS tumor types are still lacking. We believe that our results offer opportunities to validate pathways that home in on actual targets for therapy but is outside of the scope of the current study.

11. This reviewer is a little suspicious of the stem-like cells not expressing stem-like markers (SOX2 and CD44). Is it biologically relevant or an artifact due to sample quality, demultiplexing or analysis?

While cells not expressing *SOX2* or *CD44* may be an artifact due to sample quality, it is possible that there are other non-canonical markers for stemness in the cells. While some stem-like cell types like oligodendrocyte precursors cells and radial glial cells express specific stem cell markers, it is evident that cell types like neural stem cells (NSCs) do not seem to directly express well known markers like *SOX2* or *CD44* [PMID: 29218080]. We were able to identify NSCs utilizing enrichment scores using many stem cell related genes and enrichment tests with NSC specific gene sets. The median stemness scores for each cell type were higher compared to more differentiated cell types like astrocytes and neurons (Supplementary Table 2). It also is possible that among the cross-sectional group of individual stem-like cells, those with sufficient *SOX2* and or *CD44* protein (or other stemness marker proteins), may not have needed to transcribe mRNA at the time when they were frozen, or limited mRNA was present and not detected.

12. Figure 3 panel C is missing from Figure 3.

Thank you pointing this out. We have revised the labels of Figure 3.

13. Overall, the authors need to be cautious to pull samples from different tumor types and grades for integrated analysis. Given there is only 1 GBM sample and 1 Schwannoma sample, some results of differential analysis need to be carefully evaluated. How is the difference in the number of nuclei controlled in the integrated analysis?

Thank you for pointing this out. We agree with the reviewer that it is important to consider that there are only 1 GBM and 1 Schwannoma sample for the differential expression analysis. There are 5795 nuclei for the glioblastoma and 2501 nuclei for the Schwannoma (Supplementary Table 1A). We thought it important to include these samples as there are limited studies that have published these results and the unit of analysis is the nuclei. We have added qualification of the single sample included for these two types in the manuscript as follows:

(pg. 15): However, comparisons with especially Schwannoma and pediatric-type high grade glioma samples need to be further studied as although thousands of nuclei were included in the analysis, they were derived from only one sample each.

14. The analytical codes used to generate the main results of the paper must be provided.

Thank you for pointing this out. We have added a GitHub link for code used in this analysis. <https://github.com/sarahmkleee/IntegrativePCNS>

REVIEWERS' COMMENTS

Reviewer #1 (Remarks to the Author):

The authors have addressed all the concerns.

Reviewer #2 (Remarks to the Author):

In general, the authors have effectively tackled the reservations I previously had.

Please make sure to revise the following sentence on UMAP to properly reflect the correct usage and interpretation of UMAP: “We utilized uniform manifold approximation and projection (UMAP), a dimension reduction strategy, to identify nuclei most similar to each other based on gene expression levels.”.

UMAP simply provides a network/distances to help identify similarities. However, the nuclei most similar to each other is identified by clustering based on gene expression similarities (not using UMAP).

REVIEWERS' COMMENTS

Reviewer #1 (Remarks to the Author):

The authors have addressed all the concerns.

Reviewer #2 (Remarks to the Author):

In general, the authors have effectively tackled the reservations I previously had.

Please make sure to revise the following sentence on UMAP to properly reflect the correct usage and interpretation of UMAP: “We utilized uniform manifold approximation and projection (UMAP), a dimension reduction strategy, to identify nuclei most similar to each other based on gene expression levels.”.

UMAP simply provides a network/distances to help identify similarities. However, the nuclei most similar to each other is identified by clustering based on gene expression similarities (not using UMAP).

The above sentence has been revised to the following:

(pg. 6): We utilized unsupervised clustering from components of uniform manifold approximation and projection (UMAP), a dimension reduction strategy, to identify nuclei most similar to each other based on gene expression levels.